# p62 filaments capture and present ubiquitinated cargos for autophagy

Gabriele Zaffagnini[1], Adriana Savova[1], Alberto Danieli[1], Julia Romanov[1], Shirley Tremel[2,†], Michael Ebner[3,‡], Thomas Peterbauer[1], Martin Sztacho[1,§], Riccardo Trapannone[1], Abul K Tarafder[2,¶], Carsten Sachse[2] ⅅ & Sascha Martens[1,*] ⅅ

## Abstract

The removal of misfolded, ubiquitinated proteins is an essential part of the protein quality control. The ubiquitin-proteasome system (UPS) and autophagy are two interconnected pathways that mediate the degradation of such proteins. During autophagy, ubiquitinated proteins are clustered in a p62-dependent manner and are subsequently engulfed by autophagosomes. However, the nature of the protein substrates targeted for autophagy is unclear. Here, we developed a reconstituted system using purified components and show that p62 and ubiquitinated proteins spontaneously coalesce into larger clusters. Efficient cluster formation requires substrates modified with at least two ubiquitin chains longer than three moieties and is based on p62 filaments cross-linked by the substrates. The reaction is inhibited by free ubiquitin, K48-, and K63-linked ubiquitin chains, as well as by the autophagosomal marker LC3B, suggesting a tight cross talk with general proteostasis and autophagosome formation. Our study provides mechanistic insights on how substrates are channeled into autophagy.

**Keywords** aggrephagy; cargo receptor; phase transition; quality control; selective autophagy
**Subject Categories** Autophagy & Cell Death; Post-translational Modifications, Proteolysis & Proteomics
**The EMBO Journal (2018) 37: e98308**

## Introduction

Protein aggregation is a major threat to cellular homeostasis. The accumulation of misfolded aggregated proteins is associated with many pathological conditions, including several neurodegenerative diseases (Menzies *et al*, 2015). In eukaryotic cells, two interconnected systems remove misfolded proteins, the ubiquitin-proteasome system (UPS) and macroautophagy (hereafter autophagy; Dikic, 2017). In the UPS, misfolded proteins are marked for degradation by the attachment of poly-ubiquitin chains to lysine (K) residues and are subsequently recognized by the proteasome, where they are unfolded, de-ubiquitinated, and eventually degraded (Finley, 2009). The UPS is the main degradation route for misfolded polypeptides. However, when the UPS is impaired or its capacity is exceeded, misfolded poly-ubiquitinated proteins can accumulate forming larger structures, which subsequently become substrates (cargoes) for selective autophagy (also referred to as aggrephagy; Dikic, 2017; Galluzzi *et al*, 2017). During aggrephagy, protein cargoes are sequestered within double-membrane organelles called autophagosomes, which are marked by attachment of ubiquitin-like modifiers of the ATG8 family to the lipid phosphatidylethanolamine (Zaffagnini & Martens, 2016). Cargoes are degraded upon the fusion of autophagosomes with the lysosomal compartment (Zaffagnini & Martens, 2016).

Selective autophagy depends on cargo receptor proteins which link the cargo to the autophagosomal membrane (Stolz *et al*, 2014; Zaffagnini & Martens, 2016). The major cargo receptor for aggrephagy is p62/SQSTM1 (Stolz *et al*, 2014). Mutations in the *SQSTM1* gene have been associated with several diseases including amyotrophic lateral sclerosis (ALS), frontotemporal dementia (FD), and neurodegeneration with ataxia (Fecto *et al*, 2011; Rubino *et al*, 2012; Goode *et al*, 2016; Haack *et al*, 2016). p62 binds to ubiquitin via its C-terminal UBA domain and to ATG8 proteins, including LC3B, through an LC3-interacting region (LIR) located in an intrinsically disordered region (Fig 1A; Seibenhener *et al*, 2004; Pankiv *et al*, 2007). In addition, the N-terminal PB1 domain of p62 mediates its oligomerization into helical filaments (Lamark *et al*, 2003; Ciuffa *et al*, 2015), allowing it to bind to ubiquitin-positive cargoes and to LC3B-decorated membranes with high avidity (Wurzer *et al*, 2015).

1  Department of Biochemistry and Cell Biology, Max F. Perutz Laboratories (MFPL), Vienna Biocenter (VBC), University of Vienna, Vienna, Austria
2  Structural and Computational Biology Unit, European Molecular Biology Laboratory, Heidelberg, Germany
3  Department of Structural and Computational Biology, Max F. Perutz Laboratories (MFPL), Vienna Biocenter (VBC), University of Vienna, Vienna, Austria
*Corresponding author. Tel: +43 1 4277 52876; E-mail: sascha.martens@univie.ac.at
†Present address: MRC Laboratory of Molecular Biology, Cambridge, UK
‡Present address: Leibniz-Forschungsinstitut für Molekulare Pharmakologie, Berlin, Germany
§Present address: Institute of Molecular Genetics, Prague 4, Czech Republic
¶Present address: Sir William Dunn School of Pathology, University of Oxford, Oxford, UK

p62 links the UPS and autophagy (Demishtein *et al*, 2017; Dikic, 2017). For instance, the accumulation of protein aggregates results in an upregulation of p62, which in turn negatively affects the UPS, promoting a switch toward aggrephagy (Nagaoka *et al*, 2004; Komatsu *et al*, 2007; Korolchuk *et al*, 2010; Homma *et al*, 2014; Rusmini *et al*, 2015). p62 is also required for the nucleation of several types of aggregates containing misfolded proteins (Zatloukal *et al*, 2002; Nan *et al*, 2004; Bjorkoy *et al*, 2005; Komatsu *et al*, 2007; Pankiv *et al*, 2007; Stumptner *et al*, 2007; Kageyama *et al*, 2014; Lahiri *et al*, 2016). During this process, p62 acts together with other proteins, including ALFY, WDR81, Huntingtin, and the cargo receptor NBR1 (Kirkin *et al*, 2009; Clausen *et al*, 2010; Rui *et al*, 2015; Liu *et al*, 2017). However, it is unclear whether p62 is sufficient for cargo nucleation and, if so, what its preferred substrates are. Here, we demonstrate in a fully reconstituted system using purified components that p62 is sufficient for the clustering of ubiquitinated proteins into larger assemblies. We identify the determinants of the cargo required for recognition by p62, revealing how autophagy might be triggered by proteasomal overload. We further discover a mechanism potentially allowing the coordination of cargo nucleation and autophagosome formation.

# Results

## p62 and ubiquitin spontaneously form clusters in solution

In order to determine whether p62 is sufficient for cluster formation, we developed a fully *in vitro* reconstituted system, imaging recombinant oligomeric mCherry-p62 (Wurzer *et al*, 2015) in the presence of ubiquitinated substrates. To mimic poly-ubiquitinated proteins, we fused GST to GFP and linear tetra-ubiquitin (GST-GFP-4xUb) (Fig 1B). p62 and GST-GFP-4xUb spontaneously coalesced into μm-sized clusters (Fig 1C, Movie EV1). No cluster formation was observed when p62 or GST-GFP-4xUb was incubated alone (Fig 1C, Movie EV1). We also observed separately purified mCherry-p62 and GFP-p62 coalescing into the same structures confirming that the clusters were not pre-formed (Fig EV1A).

To quantify the clustering reaction, we determined the number and size of the clusters (Fig EV1B, Code EV1). The number of clusters rose sharply shortly after mixing p62 and GST-GFP-4xUb and subsequently gradually declined (Fig 1D and E) due to merging of the clusters, and their sedimentation. The average cluster size steadily grew over time (Fig 1F) while their estimated total volume increased rapidly and slowly declined only toward the end of the imaging period (Fig 1G). We were unable to detect the same number and size of particles in both channels, likely due to the

higher concentration of GST-GFP-4xUb, which increased the background signal (Fig EV1C). Thus, p62 and ubiquitinated proteins are sufficient to nucleate clustered structures in solution.

*In vivo* cargo nucleation by p62 requires oligomerization and ubiquitin binding, which are mediated by its N-terminal PB1 domain and C-terminal UBA domain, respectively (Fig 1A; Bjorkoy *et al*, 2005; Komatsu *et al*, 2007; Pankiv *et al*, 2007). p62 behaved similarly in our reconstituted system as mCherry-tagged p62ΔPB1 and p62ΔUBA showed little or no cluster formation (Fig 1H). The K7A/D69A mutant, which is partially deficient in oligomerization (Lamark *et al*, 2003; Wurzer *et al*, 2015), showed reduced cluster formation (Fig 1H) and the p62 phospho-mimicking mutant S403E, which has an increased affinity for ubiquitin (Matsumoto *et al*, 2011), showed increased clustering (Fig 1H). Oligomerization and ubiquitin binding of p62 were also required for the incorporation of mCherry-p62 mutants into clusters formed by wild-type GFP-p62 (Fig EV1D). Thus, p62 in our reconstituted system supports cluster formation in the same way that it does in cells.

## p62 is oligomeric *in vivo* and forms clusters with ubiquitin

To investigate the behavior of p62 *in vivo*, we tagged the endogenous p62 in human Hap1 cells using CRISPR/Cas9 (Fig 2A). StrepII-Tev-GFP-p62 (STG-p62) was expressed to similar levels than the non-tagged p62 protein (Fig EV2A). The gene-edited cells formed p62- and ubiquitin-containing puncta upon treatment with puromycin, as seen in wild-type cells (Fig 2B, arrowheads). STG-p62 cells also showed a normal p62 degradation pattern and increased LC3B lipidation upon starvation (Fig EV2A and B, respectively). We isolated STG-p62 from Hap1 cells and showed that it co-purified with known p62 interactors, including NBR1, ubiquitin, and KEAP1 (Table EV1; Seibenhener *et al*, 2004; Kirkin *et al*, 2009; Komatsu *et al*, 2010). Purified STG-p62 also formed clusters with GST-4xUb (Fig 2C). We therefore conclude that the recombinant p62 truthfully recapitulates the behavior of the endogenous protein.

To determine the native oligomerization state of p62 in cells, we performed fluorescence correlation spectroscopy (FCS) experiments with STG-p62 cells. We generated a calibration curve with tandem copies of GFP (GFP$_{1-5}$ ruler, Figs 2D and E, and EV2C and D; Pack *et al*, 2006; Vamosi *et al*, 2016). The autocorrelation curves of STG-p62 were shifted to the right in comparison with the GFP$_{1-5}$ ruler, indicating that STG-p62 is found in large complexes (Fig EV2D). To fit the curves, we needed to assume two populations characterized by distinct diffusion coefficients, which we interpolated into the calibration curve (Figs 2D and EV2E). The fast component of STG-p62 consisted of about 70% of the total (71 ± 14%) and corresponded to a predicted molecular weight of ~450 KDa (445 KDa ± 171 KDa),

---

**Figure 1. p62 and ubiquitin-positive proteins spontaneously cluster in solution.**

A   Schematic representation of the p62 architecture. Mutations tested in this study are indicated. PB1: Phox and Bem1p domain, ZZ: Zinc finger, LIR: LC3-interacting region, UBA: ubiquitin-associated domain.

B   Schematic representation of the constructs employed in the clustering assay.

C   Representative micrographs of a clustering assay with the proteins indicated on the left. GFP and mCherry fluorescence are consistently displayed in cyan and magenta, respectively.

D–G   Quantification of particle number, size, and estimated volume determined from the data shown in (C).

H   Quantification of cluster formation by the indicated p62 mutants.

Data information: For all the graphs, averages and SDs from at least three independent replicates are shown.

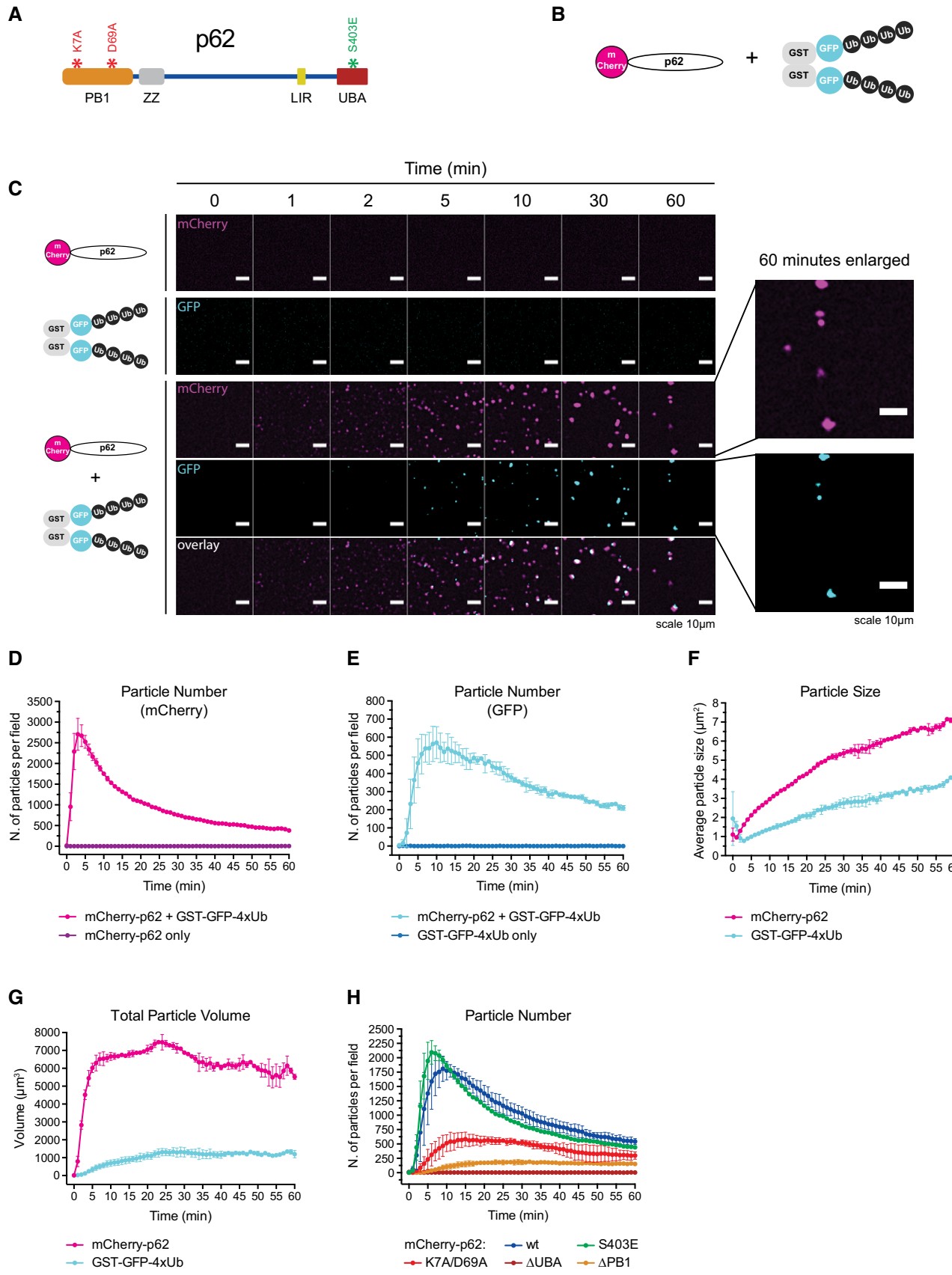

**Figure 1.**

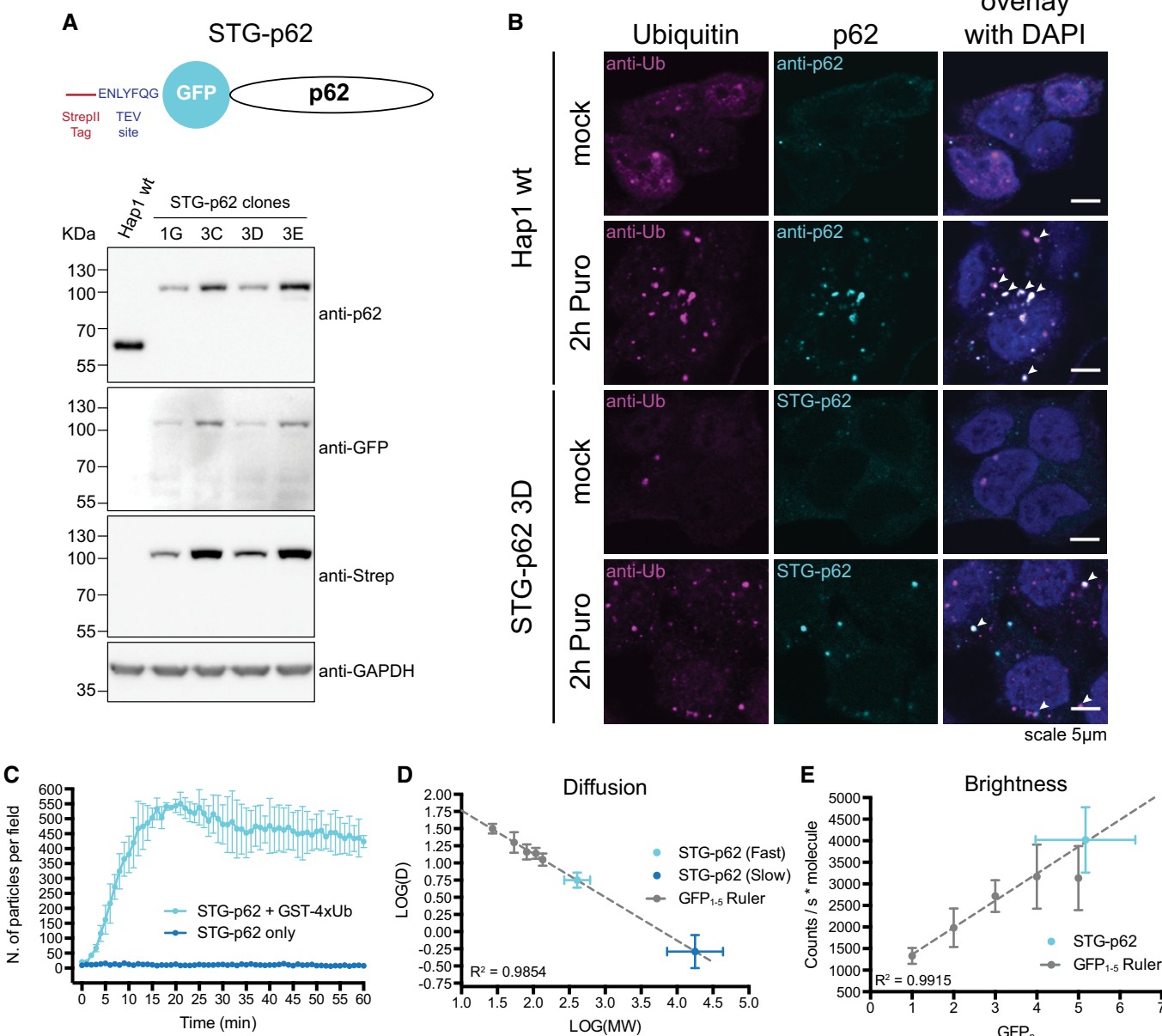

**Figure 2. Endogenously tagged p62 is oligomeric and forms clusters with ubiquitin.**

A    Top: schematic representation of the STG-p62 construct. Bottom: characterization of several STG-p62 clones in comparison with parental Hap1 cells.

B    Wild-type or STG-p62 Hap1 cells were mock- or puromycin-treated for 2 h and stained for ubiquitin. The endogenous p62 was stained with an anti-p62 antibody in the parental Hap1 cells, while the fluorescence of the GFP tag was recorded in STG-p62 cells. Arrowheads indicate colocalizing puncta.

C    Purified STG-p62 was tested for cluster formation in the presence of 20 µM GST-4xUb. Averages and SDs of three independent replicates are shown for the +GST-4xUb sample and of two for STG-p62 alone.

D, E    FCS analysis of diffusion (D) and brightness (E) of the GFP$_{1-5}$ ruler and of STG-p62 in the cytoplasm of Hap1 cells. Dashed lines represent the linear regression of the GFP$_{1-5}$ ruler data. Regression coefficients are indicated. For all data points, averages and SDs (on both axes for STG-p62) are shown (see Materials and Methods for a list of the number of measurements performed for each construct). MW: molecular weight. D: diffusion coefficient (µm²/s).

Source data are available online for this figure.

while the slow component corresponded to ~27 MDa, which might correspond to a fraction of p62 localized to autophagosomal membranes and/or forming filamentous aggregates (Stumptner et al, 2007; Ciuffa et al, 2015). Since the molecular weight of a single STG-p62 molecule is 77.5 KDa, the 450 KDa complexes could be oligomers of STG-p62 consisting of about six subunits (5.75 ± 2.22), or could contain fewer subunits of STG-p62 in complex with other proteins. We therefore analyzed the molecular brightness from the FCS data (Fig 2E). The brightness of STG-p62 corresponded to about five EGFP molecules (5.2 ± 1.2), which is very close to the diffusion measurement (Fig 2D and E). We conclude that endogenous p62 exists in oligomers and larger structures *in vivo*.

## Cluster formation depends on ubiquitin chain type, concentration, length, and clustering

To characterize the requirements for p62-mediated clustering, we determined how long the ubiquitin chains attached to GST must be to induce cluster formation. We linked linear ubiquitin chains of different lengths to GST and tested their capacity to form clusters with p62 (Fig 3A). Under the conditions tested, GST-4xUb resulted in a robust clustering reaction. GST-3xUb triggered only a weak response, while GST-2xUb showed no detectible cluster formation. Thus, assembly strongly depends on the length of the substrate-attached ubiquitin chains (Fig 3A and B). We next tested whether the ubiquitin chains need to be attached to a substrate to trigger cluster formation and added biotinylated M1-linked 4xUb to p62 (Fig 3C and D). Upon addition of streptavidin, we detected robust cluster formation (Fig 3C and D). Streptavidin alone and biotiny-lated 4xUb in the absence of streptavidin did not induce any reaction, suggesting that free chains fail to initiate it. Consistently, when we added non-biotinylated M1-linked 4xUb and streptavidin simultaneously to p62, no cluster formation was observed (Fig 3C and D).

GST is dimeric and streptavidin is a tetramer, and therefore, all the substrates tested above that triggered cluster formation carried at least two ubiquitin chains. We tested whether one ubiquitin chain attached to a substrate is sufficient to initiate the reaction by fusing M1-linked 4xUb to GFP (GFP-4xUb). GFP-4xUb did not support cluster formation, suggesting that at least two ubiquitin chains (of more than three moieties) on the same substrate are required for an efficient reaction (Fig 3E). The clustering reaction was also highly sensitive to the concentration of GST-GFP-4xUb but not of p62 (Figs 3F and EV3A). Thus, p62-dependent clustering responds rapidly to small increases in the concentration of highly ubiquitinated proteins, as expected for a system that detects proteasomal overload.

In the UPS, K48-linked ubiquitin chains are a canonical proteasomal degradation signal, although other chain types can be employed (Yau & Rape, 2016; Yau *et al*, 2017). p62 binds preferentially to K63-linked ubiquitin chains over K48-linked chains, and its deficiency leads to an accumulation of K63-linked ubiquitin (Seibenhener *et al*, 2004; Babu *et al*, 2005; Wooten *et al*, 2008;

Wurzer *et al*, 2015). To determine whether certain chain types are preferred substrates for the p62 cluster formation, we generated free K48- and K63-linked ubiquitin chains (Dong *et al*, 2011), neither of which supported cluster formation by p62 (Fig EV3B) under the conditions tested. We then generated GST-tagged K48- and K63-linked ubiquitin chains employing a single ubiquitin lacking the two C-terminal glycine residues fused to GST (GST-1xUbΔGG) as a substrate for ubiquitin chain elongation (Fig EV3C; Dong *et al*, 2011). Since we were unable to isolate GST-tagged chains of a precise length (Fig EV3C), we mixed GST-4xUb, GST-3xUb, and GST-2xUb at a molar ratio of 1:1:1 as a reference (Fig 3G). All chains triggered cluster formation (Fig 3G). However, while K63-linked ubiquitin chains elicited a response similar to linear ubiquitin, K48-linked chains were a less efficient substrate (Fig 3G).

In the UPS, substrate de-ubiquitination at the proteasome is essential for its efficient degradation (Finley, 2009). Two proteasome-associated de-ubiquitinases (DUBs) release single ubiquitin molecules, while the third cleaves ubiquitin chains *en bloc* (Lee *et al*, 2011). Free ubiquitin chains are subsequently further cleaved to replenish the pool of free ubiquitin (Kimura *et al*, 2009). Thus, the occurrence of unanchored ubiquitin chains and a high concentration of free ubiquitin correlate with active proteasomes (Kaiser *et al*, 2011). We asked whether the presence of free ubiquitin chains could influence cluster formation by p62 by adding free linear, K48-, and K63-linked ubiquitin chains to p62 + GST-4xUb clustering reactions (Fig 3H and I). Free K48-linked ubiquitin chains inhibited the induction of clusters when pre-mixed with p62, while linear and K63-linked chains reduced the reaction under the conditions tested (Fig 3H). A similar effect was also obtained with high concentrations of mono-ubiquitin (Fig EV3D). K48-linked ubiquitin chains also strongly inhibited the formation of clusters when they were added to ongoing clustering reactions (Fig 3I). Free ubiquitin chains, and in particular K48-linked chains, have been shown to disassemble PB1-dependent p62 oligomers (Ciuffa *et al*, 2015; Wurzer *et al*, 2015). We therefore asked whether we could render the cluster formation reaction resistant to the inhibition by K48- and K63-linked chains by replacing the PB1 domain with an inducible oligomerization domain. To this end, we replaced the p62 PB1 domain with two FK506 binding protein (FKBP) domains (Itakura & Mizushima, 2011). Upon addition of the homodimerizer AP20187,

---

**Figure 3.  p62-dependent cluster formation is triggered by poly-ubiquitinated proteins.**

A    Representative micrographs of cluster formation assays conducted with GST-tagged linear tetra-, tri- or di-ubiquitin. The total amount of ubiquitin moieties was kept identical in each condition. Scale bars: 10 μm.

B    Quantification of the data shown in (A).

C    Quantification of cluster formation assays conducted with the indicated proteins. mCherry-p62 was pre-incubated with the indicated 4xUb variants, followed by the addition of streptavidin immediately before imaging.

D    Representative micrographs of the indicated samples taken 60 min after addition of streptavidin.

E    Quantification of cluster formation assays conducted with the indicated proteins.

F    Quantification of cluster formation assays conducted with 2 μM mCherry-p62 incubated with the indicated concentration of GST-GFP-4xUb. Box: the highest particle number per field detected in each sample was plotted against the relative GST-GFP-4xUb concentration. Data points were fitted to a sigmoidal curve. The regression coefficient is indicated.

G    Quantification of cluster formation assays conducted with GFP-p62 and the indicated GST-tagged ubiquitin chains.

H, I  Quantification of cluster formation assays performed with mCherry-p62 and GST-4xUb in the presence of the indicated free ubiquitin chains. Ubiquitin chains were either pre-mixed with p62 before the addition of GST-4xUb (H), or added 3 min after the addition of GST-4xUb to p62 (I).

J    Quantification of cluster formation assays conducted with mCherry-2xFKBP-p62 in the presence of AP20187, GST-4xUB, and the indicated Ub chains.

K    Representative micrographs of glutathione beads coated with the indicated GST-tagged proteins and incubated with the indicated mCherry-p62 variants.

Data information: In all pictures, the brightness was adjusted to highlight the profile of the beads. For all graphs, averages and SDs from at least three independent replicates are shown.

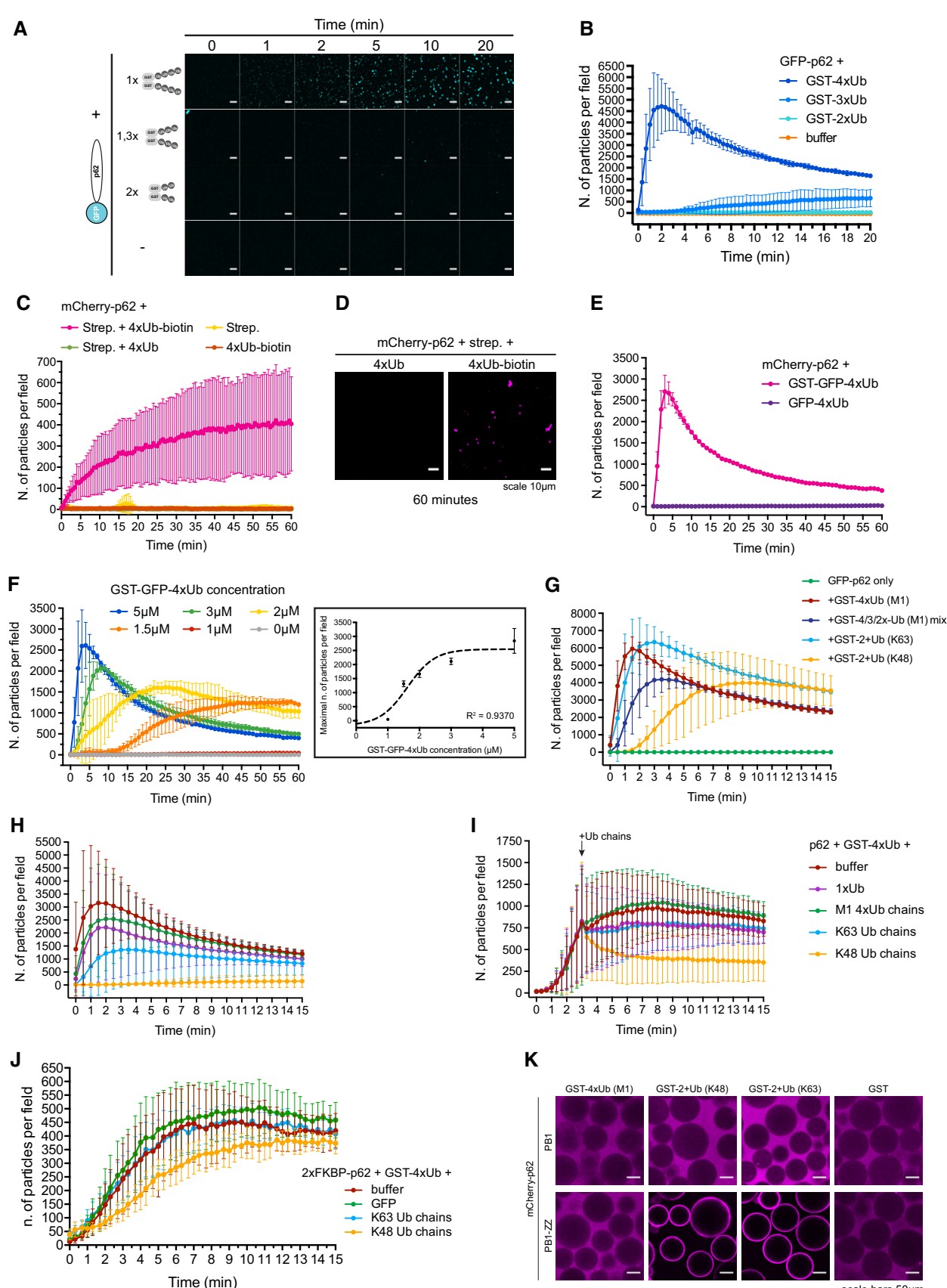

**Figure 3.**

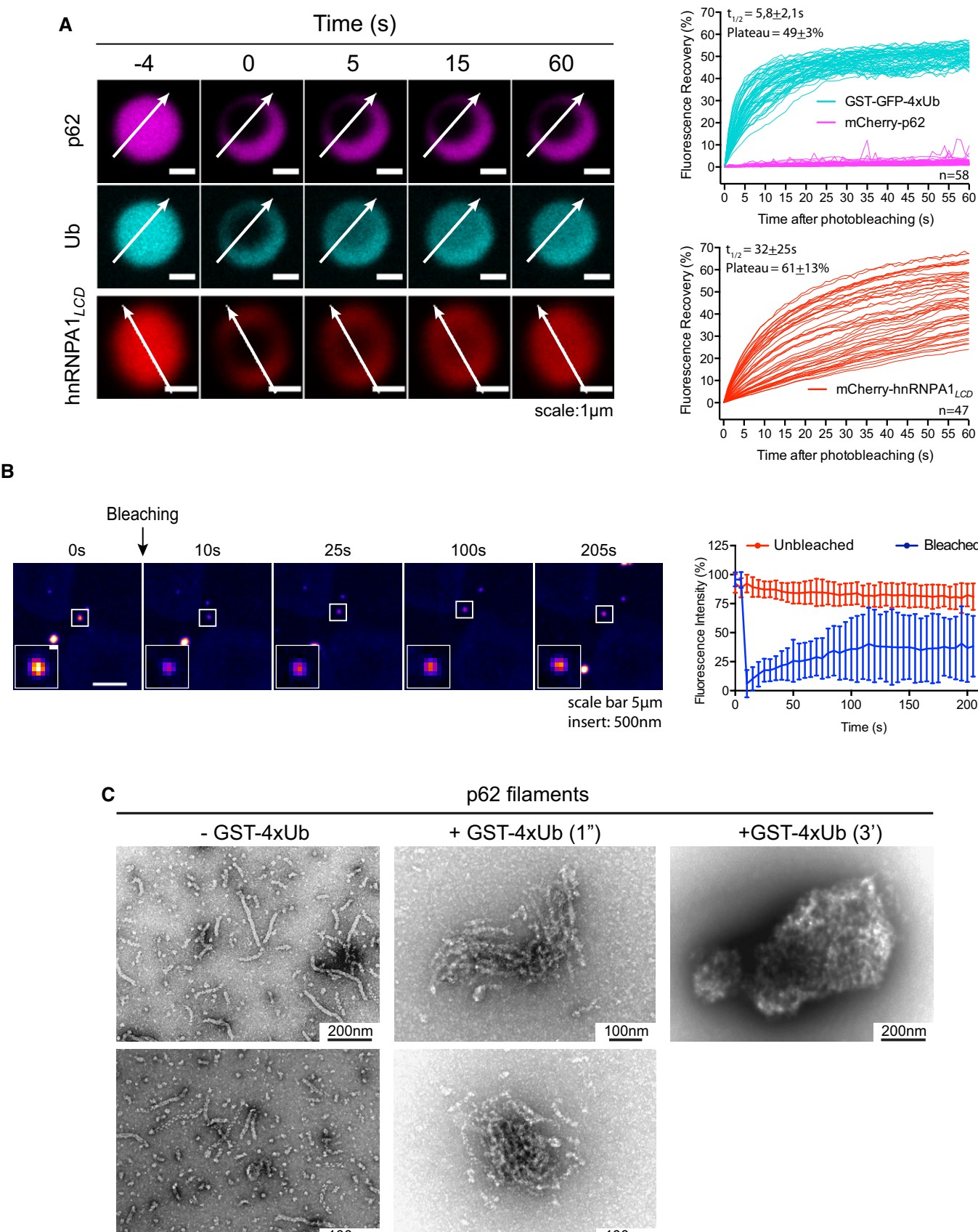

**Figure 4.**

**Figure 4.  p62 and ubiquitin show different motilities within the clusters.**

A   mCherry-p62/GST-GFP-4xUb clusters were photobleached, and fluorescence recovery in the bleached area was monitored. The mCherry-tagged low-complexity domain (LCD) of hnRNPA1 (hnRNPA1$_{LCD}$) was included as a reference for a liquid droplet-forming protein. Left: representative micrographs of bleached particles. White arrows indicate the position and orientation of the kymographs shown in Fig EV4A. Right: quantification of fluorescence recovery. Individual traces of all particles analyzed from three independent replicates are displayed.

B   p62 puncta in STG-p62 cells were photobleached, and the fluorescence recovery was monitored. Left: representative micrographs of bleached cells. White squares indicate the bleached puncta. Insert: magnification of the bleached puncta. Right: quantification of fluorescence recovery. Averages and SDs of 28 (unbleached) and 29 (bleached) puncta from three independent experiments are shown.

C   Representative electron micrographs of negatively stained p62 filaments incubated in the presence or absence of GST-4xUb for the indicated times.

the protein shifted toward a higher apparent molecular weight in size-exclusion chromatography and sustained the clustering of GST-4xUb more robustly than in the absence of AP20187 (Fig EV3E and F). The addition of unanchored K48- and K63-linked Ub chains to a clustering reaction conducted with mCherry-2xFKPB-p62 exerted only a minor inhibitory effect for the K48-linked chains (Fig 3J). Thus, unanchored K48- and K63-linked chains impact cluster formation in a PB1 domain-dependent manner. To obtain further insights into the molecular mechanism underlying this effect, we turned our attention to the Zinc finger (ZZ) domain of p62 (Fig 1A) since many Zinc fingers are known to bind ubiquitin (Dikic *et al*, 2009). Indeed, a construct containing the PB1-ZZ domain, but not the isolated PB1 domain, bound robustly to K48- and K63-linked chains (Fig 3K). Linear ubiquitin chains were not bound by the PB1-ZZ protein under the conditions tested (Fig 3K). Thus, ubiquitin binding to the ZZ domain might sterically interfere with the PB1 domain-mediated oligomerization.

## p62 clusters arise from cross-linked p62 filaments

Many subcellular structures (referred to as liquid droplets) are thought to arise from phase separation reactions driven by multiple low-affinity but high-avidity interactions (Li *et al*, 2012). p62 binds to locally concentrated ubiquitin with low affinity but high avidity, due to its multimeric nature (Long *et al*, 2008; Ciuffa *et al*, 2015; Wurzer *et al*, 2015). In addition, our results suggest that multiple p62–ubiquitin interactions are required for cluster formation (Figs 1H and 3A–E). To test whether the p62–ubiquitin clusters behave as liquid droplets, we formed clusters with mCherry-p62 and GST-GFP-4xUb, and performed fluorescence recovery after photobleaching (FRAP) experiments (Fig 4A, Movie EV2). Liquid droplet-forming proteins such as the low-complexity domain of hnRNPA1 (hnRNPA1$_{LCD}$) display a rapid recover of the fluorescence due to rapid exchange with the material in solution (Figs 4A and EV4A; Li *et al*, 2012; Lin *et al*, 2015; Molliex *et al*, 2015). However, mCherry-p62 showed almost no recovery, indicating that it is rather stably sequestered within the structures (Figs 4A and EV4A, Movie EV2). In stark contrast, GST-GFP-4xUb showed a rapid recovery showing that the protein is highly mobile within the clusters (Figs 4A and EV4A, Movie EV2).

To confirm the result that p62 showed little intra-cluster mobility, we performed structured illumination microscopy (SIM; Fig EV4B). We formed clusters with unlabeled GST-4xUb and mCherry-p62 and then added GFP-p62. GFP-p62 was observed as a layer on the surface of red clusters, and no intermixing of mCherry-p62 and GFP-p62 was detected (Fig EV4B). We obtained the complementary result when we reversed the order of addition

(Fig EV4B). As a control, we pre-mixed mCherry-p62 and GFP-p62 prior to GST-4xUb addition and found the fluorescence of mCherry and GFP to be largely, but not completely, overlapped (Fig EV4B). We also mixed separated pre-formed mCherry-p62- and GFP-p62-containing clusters and found they attached to each other, but no intermixing was observed (Fig EV4B).

We next asked how dynamically p62 exchanges with the puncta observed in cells. We therefore performed FRAP experiments with p62 puncta similar to previous studies (Matsumoto *et al*, 2011; Lee *et al*, 2017), but for the first time employing cells with endogenously GFP-tagged p62 (Figs 2A and B, and 4B). In general, p62 showed higher recovery in cells than *in vitro*, but the recovery of the puncta was slow and plateaued below 40% (Fig 4A and B). We conclude that although more mobile in cells than *in vitro*, p62 is rather stably associated with the punctate structures in cells.

Next, we performed negative staining electron microscopy (EM) of the p62–ubiquitin clusters. p62 alone was observed as 15-nm-wide filaments (Fig 4C, left; Ciuffa *et al*, 2015). Upon co-incubation of p62 with GST-4xUb, the filaments rearranged into side-by-side assemblies (Fig 4C, middle), followed by the formation of larger clusters of 0.5–2.0 μm in size (Fig 4C, right). Taken together, these results suggest that p62 clusters are formed by p62 filaments cross-linked by poly-ubiquitinated substrates, within which the ubiquitin-positive substrates show high mobility, while the p62 filaments remain relatively fixed in space.

## NBR1 directly cooperates with p62 to render clustering more efficient

p62 interacts with the autophagy receptor NBR1, and the two receptors have been suggested to cooperate during the cargo nucleation of ubiquitinated proteins (Lamark *et al*, 2003; Kirkin *et al*, 2009). We tested whether the addition of NBR1 would directly enhance the ability of p62 to form clusters. Purified GFP-NBR1 was robustly and specifically recruited to both GST-LC3B- and GST-4xUb-coated beads (Fig 5A). GFP-NBR1 also directly interacted with p62 since wild-type mCherry-p62 and the isolated PB1 domain of p62 bound to GFP-NBR1-coated beads (Fig 5B and C). In contrast, p62ΔPB1 and the p62-NBR1 chimera, in which the PB1 domain of p62 was replaced with that of NBR1 (Wurzer *et al*, 2015), were not recruited (Fig 5B and C).

To test whether NBR1 affects aggregation by p62, we added an equimolar amount of GFP-NBR1 to mCherry-p62 + GST-4xUb aggregation reactions (Fig 5D and E). GFP-NBR1 had a robust stimulatory effect on the clustering reaction (Fig 5E), with the clusters positive for both p62 and NBR1 (Fig 5D). The stimulatory effect of NBR1 on p62 clustering is not due to the capacity of NBR1 to form clusters

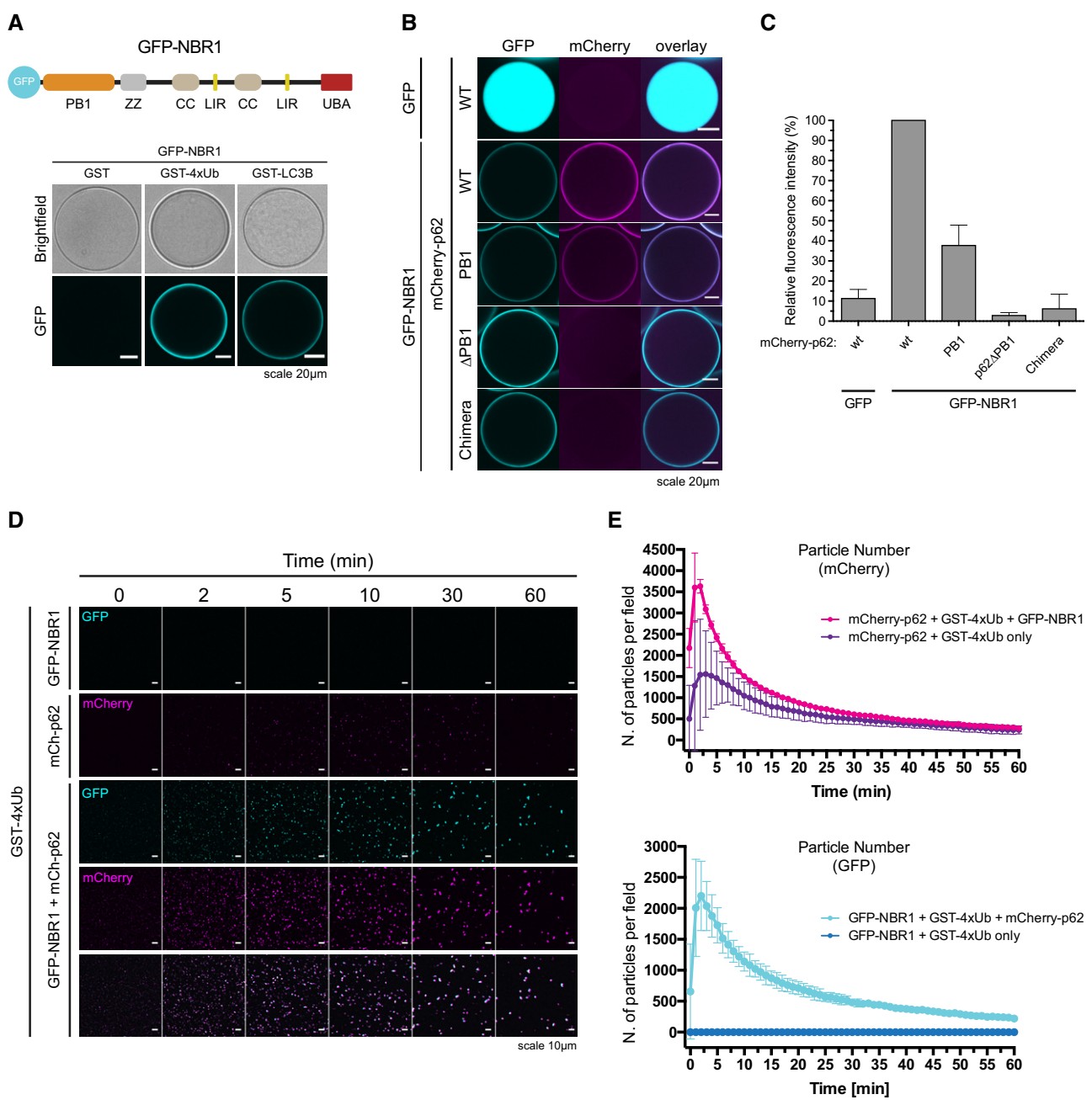

**Figure 5. NBR1 cooperates with p62.**

A    Top: schematic representation of the domain architecture of the GFP-NBR1 construct. PB1: Phox and Bem1p domain, ZZ: Zinc finger, CC: coiled coil, LIR: LC3-interacting region, UBA: ubiquitin-associated domain. Bottom: GFP-NBR1 was recruited to glutathione beads coated with the indicated GST fusion proteins and imaged by spinning disk microscopy. Representative micrographs of at least 70 beads imaged per sample from two independent experiments.

B, C    mCherry-p62 variants were recruited to GFP- or GFP-NBR1-coated beads and imaged as in (A). (B) Representative micrographs of at least 70 beads per sample from three independent replicates. (C) Quantification of the data shown in (B). Data were normalized to p62 wild-type binding to GFP-NBR1 within each replicate. Averages and SDs of three independent experiments are shown.

D, E    2 μM mCherry-p62 and/or GFP-NBR1 were incubated in the presence of 5 μM GST-4xUb. Representative micrographs (D) and quantification of particle number for each channel (E) are shown. Averages and SDs from at least three independent experiments are shown.

with ubiquitin by itself (Fig EV5A). In contrast, optineurin (OPTN), another ubiquitin-binding cargo receptor that is known to act independently of p62, was not recruited to p62 clusters and did not influence their formation (Fig EV5B and C) (Wild *et al*, 2011). Thus, NBR1, unlike OPTN, directly cooperates with p62 for the clustering of ubiquitin-positive proteins.

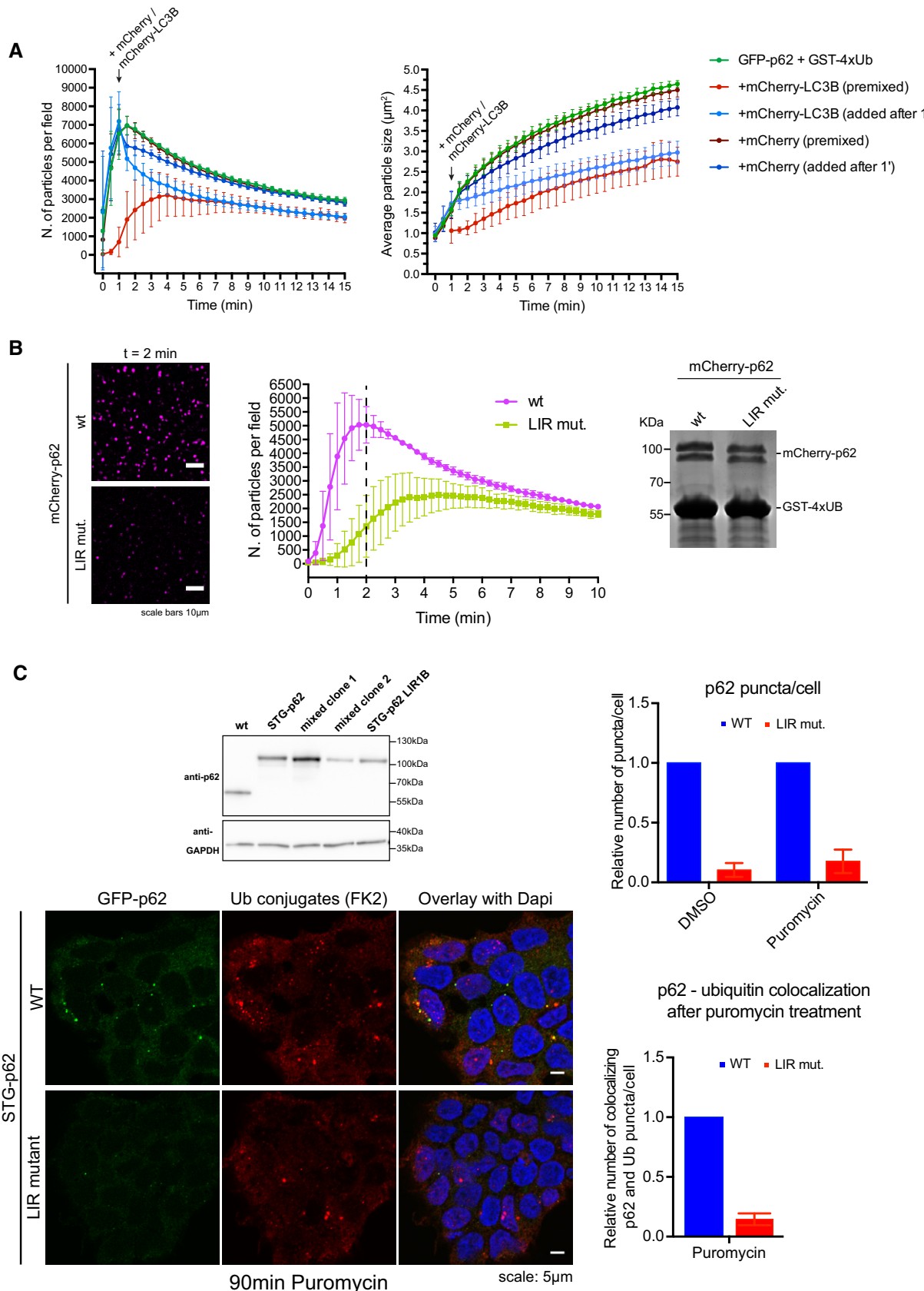

**Figure 6.**

**Figure 6.  LIR-mediated cross talk between cluster formation and the autophagy machinery.**

A  Cluster formation assays were conducted in the presence or absence of 20 μM mCherry-LC3B or mCherry, either pre-mixed with p62 or added 1 min after the addition of GST-4xUb. Quantification of particle number (left) and size (right) are shown.

B  Cluster formation assays were conducted with mCherry-p62 wt or LIR mutant in the presence of GST-4xUb. Left: representative images of the samples two minutes after the addition of GST-4xUb. Middle: quantification of particle count. The dashed line indicates the time point of images displayed on the left. Right: input gel of the samples after reaction. The relative position of marker bands is indicated.

C  Top left: Western blots of lysates of CRISPR/Cas9 genome-edited STG-p62 wild-type and LIR mutant cells. Clone Lir1B was used for further experiments shown here. Bottom left: Representative pictures of STG-p62 WT or LIR mutant Hap1 cells treated with puromycin for 90 min and stained with anti-Ub conjugates antibody (FK2). Right: quantification of cytoplasmic p62 puncta per cell (top), and of their co-localization with ubiquitin upon puromycin treatment (bottom). In total, at least 1,440 cells were counted per condition.

Data information: For all graphs, averages and SDs from at least three independent replicates are shown.
Source data are available online for this figure.

### p62 clusters interact and crosstalk with the autophagic machinery

During aggrephagy, p62 interacts with ATG8-family proteins such as LC3B on the autophagosomal membrane (Pankiv *et al*, 2007; Ichimura *et al*, 2008; Zaffagnini & Martens, 2016). We found that *in vitro* pre-formed p62 clusters have the ability to recruit LC3B in a LIR-dependent manner (Fig EV6A). Surprisingly, pre-incubation of p62 with LC3B reduced the number and size of the clusters formed upon addition of GST-4xUb (Fig 6A). In cells, the nascent autophagosomal membrane becomes progressively decorated with ATG8-family proteins and therefore their local concentration increases. We observed that addition of mCherry-LC3B, but not of mCherry, to growing clusters resulted in a rapid slowdown of their formation and growth (Fig 6A). This effect was specifically caused by LC3B binding to the LIR motif of p62, since the addition of LC3B to the LIR mutant of p62 elicited only a minor slowdown of clustering (Fig EV6B).

We next sought to determine the molecular basis underlying the LC3B-mediated inhibition of cluster formation. Unlike ubiquitin chains, the inhibitory effect of LC3B was not dependent on the presence of the PB1 domain, since clustering of the artificially oligomerizing mCherry-2xFKBP-p62 construct was also sensitive to the addition of LC3B (Figs 3J and EV6C). This was in line with the fact that the oligomerization state of p62 LIR mutant was identical to that of the wild-type protein (Wurzer *et al*, 2015). We reasoned that the inhibitory effect of LC3B may be due to steric interference with the clustering reaction due to its binding to the LIR motif in p62. Alternatively, the LIR motif of p62 may be directly involved in cluster formation and LC3B binding might mask this function. To test these models, we determined the ability of p62 with a mutated LIR motif (DDDWTHL to AAAATHL) to form clusters in the presence of GST-4xUb. Cluster formation was substantially reduced for the LIR mutant (Fig 6B), implying a direct involvement of the LIR motif in the clustering reaction. Furthermore, the p62 LIR mutant also showed reduced recruitment to pre-assembled p62–ubiquitin clusters (Fig EV6D).

The LIR motif of p62 did not mediate detectable binding to p62 itself, since full-length GFP-p62 was not recruited to beads coated with the GST-tagged LIR region (Fig EV6E). In addition, the LIR motif did not mediate detectable direct binding to ubiquitin, since a p62 variant lacking the C-terminal UBA domain but containing even four LIR motifs in tandem also failed to bind to GST-4x(M1)Ub-coated beads (Fig EV6F). In summary, while the molecular basis for the promoting role of the LIR motif in the clustering reaction remains to be elucidated, it is unlikely to act via direct ubiquitin binding.

To test whether the LIR motif would also promote p62–ubiquitin cluster formation in cells, we introduced the LIR mutation into the STG-p62 cells via CRISPR/Cas9-mediated gene editing (Fig 6C). Consistent with the results *in vitro* suggesting a cluster promoting effect of the LIR motif, STG-p62 LIR mutant cells accumulated less p62-positive puncta than STG-p62 wild-type cells, and after puromycin treatment, the number of p62 puncta colocalizing with ubiquitin was reduced in LIR mutant cells compared to the cells expressing wild-type STG-p62 (Fig 6C). Together, these results suggest the existence of a cross talk between clustering and the recruitment of the autophagy machinery during autophagosome formation.

## Discussion

The clearance of misfolded proteins is crucial for the maintenance of intracellular homeostasis. Misfolded, ubiquitinated proteins are normally degraded via the UPS, but when this is overloaded or malfunctioning, these proteins accumulate and are subsequently sequestered into larger structures that become targets for aggrephagy (Kageyama *et al*, 2014; Demishtein *et al*, 2017; Dikic, 2017; Galluzzi *et al*, 2017). The autophagic cargo receptor p62 is required for the nucleation of such structures (Bjorkoy *et al*, 2005; Komatsu *et al*, 2007; Pankiv *et al*, 2007; Kageyama *et al*, 2014). Here, we have found that p62 is sufficient to cluster ubiquitinated proteins. p62 efficiently targets proteins carrying two or more ubiquitin chains consisting of at least three ubiquitin moieties. In cells, multiple chains may also be provided by different misfolded polypeptides modified with one chain each, once they aggregate. It is also possible that single longer ubiquitin chains can be targeted by p62. Substrate-attached linear, K48-, and K63-linked ubiquitin chains were all able to trigger cluster formation by p62. K63-linked chains were more potent than K48-linked chains under the conditions tested, implying that clustering in the absence of proteasomal inhibition could be specifically triggered by K63-linked chains, while substrates modified with K48-linked chains might only be accepted at higher concentrations resulting from a dysfunctional UPS. Alternatively, some chain remodeling step may be additionally required to trigger assembly *in vivo* (Papadopoulos *et al*, 2017). Other ubiquitin chain types such as K11- and K33-linked ubiquitin chains may also be able to trigger cluster formation (Nibe *et al*, 2017; Yau *et al*, 2017).

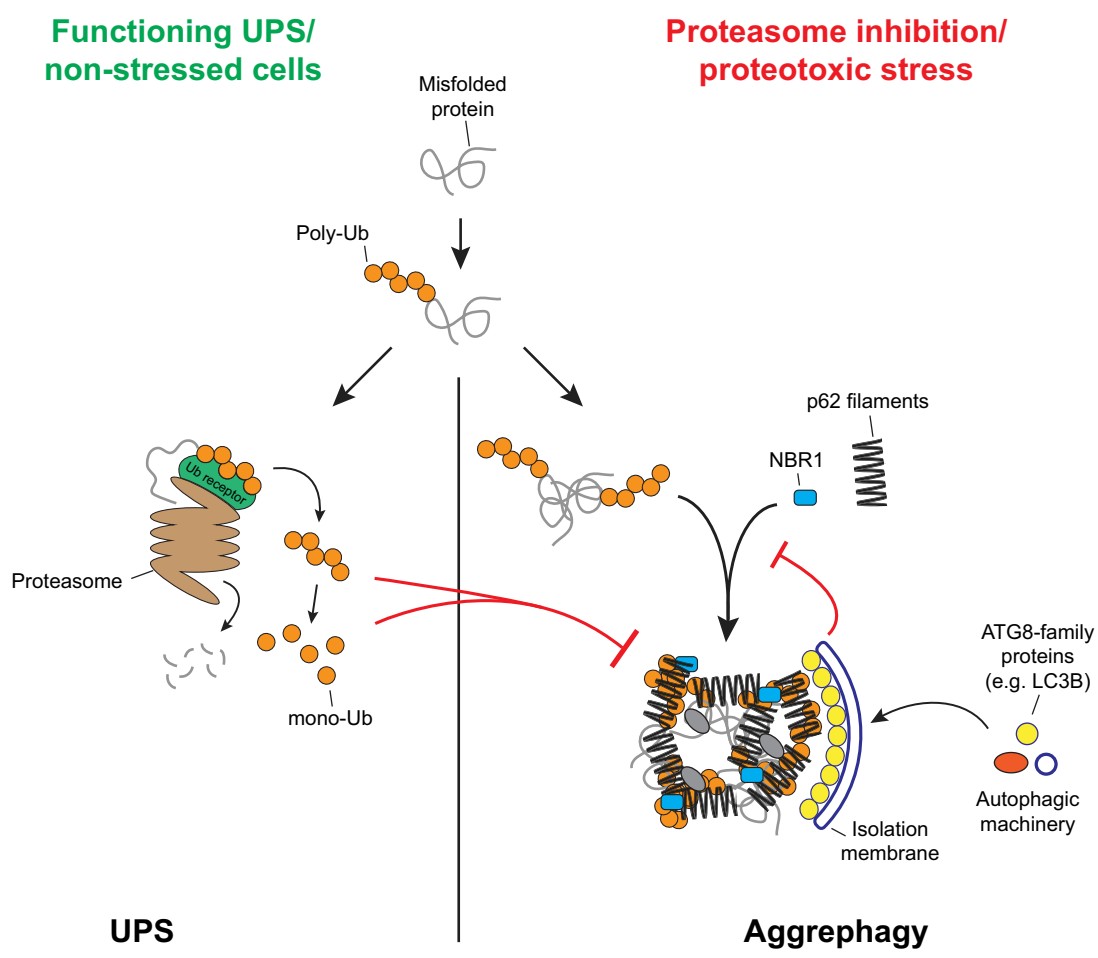

**Figure 7. p62-dependent aggrephagy compensates for proteasomal deficiency or overload.**
Model for p62-dependent cargo nucleation and autophagic degradation of poly-ubiquitinated proteins, and its cross talk with the UPS.

We also found that free mono-ubiquitin at high concentrations and unanchored ubiquitin chains, in particular K48-linked and to a lesser extent K63-linked chains, inhibit clustering. Unanchored ubiquitin chains, and most potently K48-linked chains, are known to disassemble p62 oligomers (Ciuffa *et al*, 2015; Wurzer *et al*, 2015). While the molecular basis for this disassembly activity remains unclear, it may be related to the ubiquitin-binding activity of the ZZ domain of p62, which we uncovered here. The ZZ domain is very close to the oligomerizing PB1 domain, and ubiquitin binding to the ZZ domain may sterically interfere with oligomerization. Surprisingly, we detected robust binding of the ZZ domain to K48- and K63-linked chains but not to linear ubiquitin chains. A possible explanation for this effect is that the ZZ domain binds the N-terminus of ubiquitin, which is not available in the GST-4xUb(M1) protein we used for the binding assay. Future studies will have to dissect the structural details of this interaction.

We found that p62-dependent clustering is very sensitive to the concentration of ubiquitinated proteins, implying that it primarily occurs when poly-ubiquitinated proteins accumulate upon proteasome deficiency (Kaiser *et al*, 2011). This effect, in combination with its negative regulation by free ubiquitin and ubiquitin chains,

could also be a mechanism by which the clustering activity of p62 is coupled to proteasome-mediated proteostasis (Fig 7). Modulation of receptor oligomerization may be a more general way to regulate the choice between the UPS and autophagy (Lu *et al*, 2017).

In cells, p62 cooperates with NBR1 (Kirkin *et al*, 2009). We found that NBR1 is unable to form clusters alone with ubiquitinated proteins, but directly renders p62-mediated clustering more efficient, further lowering the threshold at which p62 gives rise to clusters. This stimulatory activity and co-recruitment to clusters may be due to both its binding to ubiquitin with a higher affinity than p62 and to its ability to bind directly to p62 via the PB1 domain (Lamark *et al*, 2003; Long *et al*, 2008; Walinda *et al*, 2014). Consistently, OPTN, which binds ubiquitin with a similar affinity to NBR1 but does not interact with p62, does not stimulate clustering and is not recruited to clusters (Nakazawa *et al*, 2016).

During selective autophagy, the cargos themselves drive the local assembly of autophagosomes and keep the membrane close to the cargo (Sawa-Makarska *et al*, 2014; Lazarou *et al*, 2015; Bertipaglia *et al*, 2016; Fracchiolla *et al*, 2016; Zaffagnini & Martens, 2016). Here, we show that within the p62 clusters *in vitro*, the p62

filaments are largely immobile. These p62 filaments may therefore provide a solid scaffold for templating the formation of an autophagosome *in vivo* (Fig 7). In cells, p62 puncta show a higher mobility than *in vitro* clusters (Fig 4A and B). This may be due to the fact that *in vivo* p62 oligomers appear to be smaller than purified p62 filaments (Figs 2D and E, and 4C), or to other cellular factors and modifications. Some of the recovery seen for the cellular puncta may also be attributed to growth from the outside of the structures (Fig EV4B).

Unlike p62, the ubiquitinated substrates in our *in vitro* aggregation reactions show high mobility. Thus, the clusters exhibit an intermediate behavior between solid and liquid phases (Wu & Fuxreiter, 2016), with a filamentous scaffold provided by the oligomeric PB1 domain and a dynamic interaction with the ubiquitinated substrates provided by the UBA domain, which is connected to the PB1 domain by a disordered region. The individual ubiquitin—UBA domain interactions may be transient, and therefore, the ubiquitinated substrates are able to move within the clusters. The larger p62 filaments on the other hand may be engaged in multiple substrate interactions at any given time and therefore be much more static. In addition, their larger size may limit their diffusion within the clusters. It is also conceivable that the ubiquitinated substrates trigger the formation of p62–p62 interactions that are subsequently independently maintained. In cells, the behavior of the ubiquitinated substrate may depend on their physicochemical properties and large, misfolded proteins are likely to show less mobility than small folded proteins.

Interestingly, the masking of the p62 LIR motif by LC3B specifically counteracts the growth of p62 clusters suggesting that the LIR motif may act as an intrinsic regulatory switch between cargo nucleation and cargo engulfment, providing directionality to the whole process (Fig 7). The molecular basis for the action of the LIR motif in the clustering reaction is currently unclear, but it might be involved in the coalescence of individual p62 filaments into larger clusters.

*In vivo,* p62 cooperates with other proteins such as ALFY, WDR81, or Huntingtin to drive cargo nucleation and clearance (Clausen *et al*, 2010; Filimonenko *et al*, 2010; Rui *et al*, 2015; Liu *et al*, 2017). It has also been reported that binding of N-terminally arginylated proteins to p62 would promote its clustering ability (Cha-Molstad *et al*, 2015, 2017). In addition, poly-ubiquitin itself might contribute to aggregation (Morimoto *et al*, 2015). Further studies will be required to elucidate these mechanisms, but the reconstituted system described here has the potential to provide both a conceptual framework and powerful model system to dissect the specific contributions of these factors and modifications to the regulation of autophagy and its cross talk with the UPS.

# Materials and Methods

### Cell lines and cell culture

Wild-type and *ATG5*$^{-/-}$ Hap1 cells were purchased from Haplogen Genomics and cultured at 37°C, 5% $CO_2$ in IMDM (Gibco) supplemented with GlutaMAX™ Supplement (Gibco), 100 units/ml penicillin, 100 μg/ml streptomycin (Gibco), and 10% FBS (Sigma). All cell lines were periodically tested for mycoplasma contamination.

**Antibodies**

| Antibody | Source | ID |
|---|---|---|
| Mouse anti-StrepII | Qiagen | 34850 |
| Mouse anti-GFP | Roche | 11814460001 |
| Mouse anti p62 lck ligand | BD Biosciences | 610832 |
| Rabbit anti-p62 | MBL | PM045 |
| Mouse anti-GAPDH | Sigma | G8795 |
| Mouse anti-LC3B | NanoTools | 0260-100 |
| Mouse anti-Ubiquitin (FK2) | Enzo | BML-PW8810 |
| Mouse anti-GST | Sigma | SAB4200237 |
| Rabbit anti-K63-linked ubiquitin | Cell Signaling | D7A11 |
| Rabbit anti-K48-linked ubiquitin | Millipore | 05-1307 |
| Goat polyclonal anti-mouse HRP | Dianova | 115-035-003 |
| Goat polyclonal anti-rabbit HRP | Dianova | 111-035-003 |
| Goat AlexaFluor® 488 anti-mouse | Invitrogen | A11008 |
| Goat AlexaFluor® 546 anti-mouse | Invitrogen | A11003 |

**Plasmids**

| Plasmid | Source | ID |
|---|---|---|
| pET-Uev1 | Cheryl Arrowsmith | Addgene 25619 |
| pET-SUMO-UbcH13 | Wiener *et al* (2012) | Addgene 51131 |
| pET-GFP-p62 | This study | SMc390 |
| pET-mCherry-p62 | Wurzer *et al* (2015) | SMc391 |
| pGEX-LC3B | Wurzer *et al* (2015) | SMc394 |
| pET-mCherry-p62 K7A/D69A | Wurzer *et al* (2015) | SMc435 |
| pGEX-1xUb | Wurzer *et al* (2015) | SMc436 |
| pET-EGFP-LC3B-His | Wurzer *et al* (2015) | SMc459 |
| pET-GFP-1xUb | Wurzer *et al* (2015) | SMc475 |
| pET-mCherry-p62 LIR mut. | Wurzer *et al* (2015) | SMc542 |
| pET-mCherry-p62 ΔPB1 | Wurzer *et al* (2015) | SMc546 |
| pET-EGFP | This study | SMc559 |
| pET-mCherry | This study | SMc560 |
| pET-mCherry-p62 Chimera | Wurzer *et al* (2015) | SMc563 |
| pET-mCherry-p62 ΔUBA | Wurzer *et al* (2015) | SMc590 |
| pET-mCherry-2xFKBP-p62 | This study | SMc689 |
| pET-mCherry-p62 S403E | This study | SMc709 |
| pET-mCherry-p62 PB1 | This study | SMc768 |
| pET-mCherry-p62 PB1-ZZ | This study | SMc769 |
| pET-mCherry-OPTN | This study | SMc797 |
| pGEX-GFP-4xUb G75A/G76A | This study | SMc831 |
| pUC19_Homology arms_STG-p62 | This study | SMc864 |
| pX462_Cas9n-2A-Puro_sgRNA A | This study | SMc865 |
| pX462_Cas9n-2A-Puro_sgRNA B | This study | SMc866 |

| Plasmid | Source | ID |
| --- | --- | --- |
| pET-GFP-p62 LIR mut. | This study | SMc875 |
| pGEX-1xUbΔGG | This study | SMc898 |
| pET-Ubiquitin | This study | SMc907 |
| pET-CDC34A | This study | SMc913 |
| pFastBac HT B StrepII-TEV-GFP-NBR1 | This study | SMc914 |
| pET-UBA1 | This study | SMc915 |
| pX458_Cas9Nuc_mCherry_p62LIRgDNA | This study | SMc932 |
| pET-mCherry-LC3B | This study | SMc948 |
| pET-mCherry-hnRNPA1 | This study | SMc1029 |

The plasmids encoding for the GFP$_{1-5}$ ruler were a kind gift of Masataka Kinjo, Hokkaido University, Japan (Pack et al, 2006). The plasmids encoding for GST-2xUb, GST-3xUb, and GST-4xUb were a kind gift from Fumiyo Ikeda, IMBA, Vienna, Austria.

**Protein expression and purification**

The construction and purification of mCherry-p62 variants, GST-LC3B, GST-2xUb, GST-4xUb, GFP-LC3B, and GFP-Ub were described previously (Wurzer et al, 2015). GFP-p62 was constructed and purified analogous to mCherry-p62. GFP-p62 LIR mut. was obtained by site-directed mutagenesis. mCherry-p62 PB1 (1–102) and mCherry-p62 PB1-ZZ (1–167) were generated by site-directed mutagenesis on the plasmid encoding for the full-length protein. mCherry-2xFKBP-p62 was generated replacing the PB1 domain of p62 (1–102) with two copies of the FK506-binding protein domain (FKBP) in tandem. His-TEV-mCherry-OPTN was cloned as following: First, the ORF encoding for human OPTN was cloned into pmCherry-C1, and then, the mCherry-OPTN fusion was cloned into pET-Duet1 with the addition of a tobacco etch virus (TEV) protease cleavage site in front of mCherry. The protein was expressed in E. coli Rosetta pLysS overnight at 18°C in the presence of 0.1 mM IPTG and purified employing a HisTrap HP 5 ml column (GE Healthcare, Buckinghamshire, UK) equilibrated in buffer A (50 mM HEPES pH 7.5, 300 mM NaCl, 10 mM imidazole, 2 mM beta-mercaptoethanol). Elution was performed via a stepwise imidazole gradient (50, 75, 100, 150, 200, and 300 mM). Protein-containing fractions were pooled and subjected to overnight cleavage with TEV protease at 4°C. The cleaved protein was applied to a Superdex 200 16/60 column (GE Healthcare) and eluted in SEC buffer (25 mM HEPES pH 7.5, 150 mM NaCl, 1 mM DTT). Fractions containing the purified proteins were pooled, concentrated, frozen in liquid nitrogen, and stored at −80°C. GST-3xUB was purified as described for GST-2xUb and GST-4xUb (Wurzer et al, 2015). GST-1xUbΔGG was generated by site-directed mutagenesis of a construct encoding for GST-1xUb (Wurzer et al, 2015), expressed in E. coli Rosetta pLysS cells for 4 h at 37°C in the presence of 1 mM IPTG, and purified as described for the parental construct (Wurzer et al, 2015). GST-GFP-4xUb was constructed as follows: A plasmid encoding for GFP-4xUb G75A/G76A was purchased from the MRC Protein Phosphorylation and Ubiquitylation Unit, University of Dundee, Dundee, UK. GFP-4xUb was subsequently cloned into pGEX-4T1 to generate GST-GFP-4xUb. The construct was expressed

in E.coli Rosetta pLysS cells overnight at 18°C in the presence of 0.1 mM IPTG. Cells were lysed as previously described (Wurzer et al, 2015). The protein was subsequently purified employing a GSTrap FF 5 ml column (GE Healthcare) equilibrated in 50 mM HEPES pH 7.5 300 mM NaCl 1 mM DTT, and eluted with 20 mM reduced L-glutathione (Sigma-Aldrich) in the same buffer. The protein was subsequently subjected to size-exclusion chromatography (SEC) employing a Superdex 200 16/60 column (GE Healthcare) equilibrated in SEC buffer (25 mM HEPES pH 7.5, 150 mM NaCl, 1 mM DTT). GFP-4xUb and 4xUb were obtained from GST-GFP-4xUb and GST-4xUb, respectively, by proteolytic cleavage with Thrombin (Serva) overnight at 4°C. The reactions were subsequently depleted of GST by incubation with glutathione Sepharose 4B beads (GE Healthcare) for 30 min at RT. The proteins in the supernatants were applied to a Superdex 75 16/60 (GE Healthcare) column equilibrated with SEC buffer.

6xHis-Strep-TEV-GFP-NBR1 was generated as follows. First, an insert containing meGFP (monomeric eGFP bearing the A206K mutation) was cloned into pcDNA3.1(+). A StrepII tag followed by a tobacco etch virus (TEV) protease cleavage site was added at the N-terminus of eGFP. The ORF of human NBR1 isoform (NM_005899) was subsequently amplified by PCR from a HeLa cDNA library and cloned in frame with eGFP. The Strep-TEV-GFP-NBR1 fusion was then subcloned into the pFastBac HT B vector with an N-terminal 6xHis tag and expressed in Sf9 insect cells using the Bac-to-Bac system (Thermo Fisher Scientific). The cells were lysed in lysis buffer (50 mM HEPES pH 7.5, 300 mM NaCl, 1 mM MgCl$_2$, benzonase, complete protease inhibitor (Roche, Basel, Switzerland), 1 mM PefaBlock (Sigma-Aldrich), 10 mM imidazole, 2 mM β-mercaptoethanol). The lysate was cleared by centrifugation, and the supernatant was applied to a HisTrap HP 5 ml column as described (Wurzer et al, 2015). The eluted protein was subjected to proteolytic cleavage with TEV and applied to a Superose6 10/300 GL column (GE Healthcare). His-TEV-mCherry-LC3B was obtained by insertion of the human LC3B ORF into pmCherry-C1 (Clontech). The mCherry-LC3B fusion was subcloned into pET-Duet1 to generate His-TEV-mCherry-LC3B. His-TEV-mCherry and His-TEV-GFP were generated by cloning the mCherry and EGFP ORFs from pmCherry-C1 and pEGFP-C1 (Clontech) into pET-Duet1, respectively. His-TEV-ubiquitin was generated by insertion of human ubiquitin into pET-Duet1. In all cases, a TEV cleavage site was included to remove the 6xHis tag provided by the vector. The proteins were expressed in E. coli Rosetta pLysS cells overnight at 18°C in the presence of 0.1 mM IPTG (or 0.5 mM for His-TEV-ubiquitin). Cells were lysed as described (Wurzer et al, 2015) and the proteins were purified using a HisTrap HP 5 ml column (GE Healthcare) as described above. Protein-containing fractions were pooled, cleaved with TEV overnight at 4°C, and subjected to size-exclusion chromatography using a Superdex 75 16/60 column (GE Healthcare) equilibrated in SEC buffer.

Plasmids encoding for the E1 and E2 enzymes required for ubiquitin chains assembly were purchased from Addgene (Entry numbers 34,965, 18,764, 25,619, 51,131 for UBA1, CDC34A, UEV1, and UBCH13, respectively). The human E1 enzyme UBA1 and E2 enzyme CDC34A were re-cloned into pET-Duet1 to generate His-UBA1 and His-CDC34A, respectively. All constructs were expressed in E. coli Rosetta pLysS cells overnight at 18°C in the presence of 0.1 mM IPTG and purified via HisTrap columns as described

above. His-UBA1 was further applied to a Superose 6 16/60 column (GE Healthcare), while His-CDC34A, His-Uev1, and His-SUMO-UBCH13 were applied to a Superdex 75 16/60 column (GE Healthcare). All size-exclusion chromatography runs were performed in SEC buffer. All the purified enzymes were supplemented with 50% glycerol, snap-frozen in liquid nitrogen and stored at −80°C. Streptavidin was purchased from Thermo Fischer Scientific, resuspended into SEC buffer, aliquoted, and flash-frozen.

Prior to all experiments, freshly thawed protein aliquots were spun at 16,000 *g* for 15 min, the supernatants were transferred to a new tube, and the protein concentrations were calculated measuring the specific absorbance at 280 nm. Extinction coefficients for each protein were calculated from the respective amino acid sequence with the ProtParam algorithm at ExPASy (http://web.expasy.org/protparam/).

### Biotinylation of linear ubiquitin chains

EZ-Link™ Sulfo-NHS-LC-Biotin was purchased from Thermo Fischer Scientific. Biotinylation of linear 4xUb was performed according to the manufacturer instructions. The required amount of Sulfo-NHS-LC-Biotin was resuspended in water immediately prior to usage and incubated with 4xUb (20-fold molar excess biotin to 4xUb) for 30 min at room temperature. Ubiquitin was subsequently purified in SEC buffer employing a Superdex 75 10/30 GL column (GE Healthcare).

### Assembly and purification of K48- and K63-linked of ubiquitin chains

The method for the assembly of ubiquitin chains with a defined linkage type was adapted from Dong *et al* (2011). Briefly, reactions containing 1 mM ubiquitin, 2 μM E1, and 20 μM E2 in 100 mM Tris pH 8, 10 mM MgCl$_2$, and 0.6 mM DTT were incubated overnight at 37°C in the presence of 10 mM ATP pH 8. His-UEV1 and SUMO-UBCH13 were purified separately and pre-mixed at a final concentration of about 600 μM each, and then diluted into the reaction. After incubation, reactions were concentrated and applied to a Superdex 75 10/300 GL column (GE Healthcare) equilibrated with SEC buffer. Fractions containing 1xUb, 2xUb, 3xUb, and ≥ 3xUb, respectively, were pooled and frozen. ≥ 3xUb chains were employed in cluster formation assays, while shorter chains were re-used in further chain elongation reactions. For the assembly of GST-tagged ubiquitin chains, 0.5 mM GST-1xUbΔGG was added to the reaction and incubated as described. Reactions were subsequently incubated with 0.25× reaction volumes of glutathione Sepharose 4B beads (GE Healthcare) for 30 min at RT, washed twice with excess of SEC buffer, and eluted with 25 mM reduced glutathione in SEC buffer. Eluates were concentrated and injected in a Superose 6 10/300 GL column (GE Healthcare) equilibrated with SEC buffer. Fractions corresponding to ≥ GST-1xUb chains were collected, flash-frozen, and subsequently employed in cluster formation assays.

### Generation of STG-p62 cell line

For N-terminal tagging of the endogenous p62 the Cas9[D10A] nickase was employed. Two guide RNAs were designed close to the start codon of p62 using the web tool crispr.mit.edu (Hsu *et al*, 2013),

and cloned individually into the sgRNA scaffold sequence of pSpCas9n(BB)-2A-Puro (PX462) V2.0 from the Zhang Lab, Addgene entry #62987 (Guide A: 5′CGCGGCTTTTGTAGCGAACGCGG; Guide B: 5′CGCCAGCTCGCCGCTCGCTATGG). For the homologous repair template, a region of ~3 Kbp of genomic DNA starting ~1.5 Kbp upstream of the START codon in the *SQSTM1* gene was cloned into pUC19. The sequence encoding for a StrepII tag followed by a TEV site and the ORF of the monomeric enhanced green fluorescent protein (eGFP[A206K]) was cloned in frame at the 5′ end of the ORF of p62. The endogenous Atg start codon was removed, and a start codon was placed in front of the StrepII tag. Poly-GlySer linkers were added between each functional moiety (StrepII, TEV, eGFP, and p62, respectively). The gRNA-specific PAM sites were removed from the homology template by site-directed mutagenesis.

Wild-type Hap1 cells were seeded on 10-cm culture dishes and transfected at 50–70% confluence (typically 24 h after seeding) using 30 μl Fugene6 (Promega) and 3.3 μg of each of the three plasmids described. Transfected cells were selected by 0.5 μg/ml puromycin treatment for 72 h, sorted by FACS for green fluorescence, and seeded into 96-well plates for single clone selection. Clones were screened for the correct integration of the STG-tag by PCR and validated by blotting against StrepII, GFP, and p62, respectively (Fig 2A). Positive clones were assessed for integrity of the autophagic flux. Clone 3D was selected for the subsequent experiments since it behaved the most similar to wild-type Hap1 cells in all the assays performed. The integrity of the genomic region of clone 3D was validated by sequencing the entire homology region and the flanking sequences.

The LIR motif of p62 (DDDWTHL, corresponding to GATGAT GACTGGACCCATCTG in the CDS) was mutated to AAAATHL (GcTGcggCcCgcGACCCATCTG) in STG-p62 3D cells via CRISPR/Cas9-mediated gene editing as follows. One guide RNA (GTTCAG GAGGAGATGATGAC) overlapping with the target sequence was chosen and cloned into a plasmid encoding for SpCas9 and mCherry as reporter (SMc932; derived from pX458_Cas9Nuc from Zhang Lab modified with mCherry). A single-stranded DNA repair template encoding for the mutated DNA plus 50- to 100-bp-long homology arms on either side was ordered from IDT (Coralville, Iowa) (CATGGTCAGGCTTGGCCTGTTGCGCGTGTCTgCTGTGTGCTCATG GTGAGTTTTGTTCCAGGAACAGATGGAGTCGGATAACTGTTCAGG AGGAGcTGcggCcCgcGACCCATCTGTCTTCAAAAGAAGTGGACCCG TCTACAGGTGAACTCCAGTC). STGp62 (clone 3D) cells at low passage number were transfected with 24.5 μl Lipofectamine 2000 (Thermo Fisher Scientific, Waltham, MA), 8.2 μg Cas9 plasmid, and 4.1 μg repair DNA per 10-cm dish. The next day, cells were FACS-sorted and cells with elevated mCherry signal were sorted to 96-well plates in groups of three cells per well to increase the survival rate. When the mixed clones were grown to visible cell clusters, genomic DNA was extracted and the presence of a mutation was tested by restriction fragment length polymorphism (RFLP) exploiting of a newly created NotI restriction site within the mutated sequence. Positive clones were expanded and single-sorted to 96-well plates. Clones deriving from single cells were expanded and tested for the presence of the mutation by sequencing a PCR product that amplified the genomic region around the LIR motif. The positive clones were also tested for the integrity of the exon structure in the cDNA and for p62 protein expression levels, integrity, and functionality as described above for the parental cell line.

## Autophagy assays

Wild-type, $ATG5^{-/-}$, and STG-p62 Hap1 cells were starved for 2 h in Earle's balanced salt solution (EBSS, Sigma) with or without addition of 400 nM Bafilomycin (Santa Cruz Biotechnology) to block lysosomal degradation, lysed, and subjected to Western blotting. For blotting against LC3B, the lysates were heated up only to 60°C for 10 min after the addition of Laemmli sample buffer.

## Cell lysis

Cells were lysed in lysis buffer [20 mM Tris pH 8, 10% glycerol, 135 mM NaCl, 0.2% Nonided P-40 Substitute (Fluka), 2.5 mM MgCl₂, DNase (Sigma), and cOmplete™ EDTA-free Protease Inhibitor Cocktail (Roche)]. Lysates were cleared by centrifugation at 16,000 $g$ for 5 min, and the supernatants were subjected to Western blotting. For the experiment shown in Fig 6C, cells were resuspended in Tris–NaCl buffer (50 mM Tris pH 8, 150 mM NaCl, 0.3 mM DTT, and cOmplete™ EDTA-free Protease Inhibitor Cocktail) and lysed by freeze-thawing. Lysates were cleared as described above.

## Immunofluorescence

For the comparison of wild-type and STG-p62 Hap1 (Fig 2B), cells were seeded on cover slides (Marienfeld Superior) and treated with 5 μg/ml puromycin or DMSO for 2 h at ~70% confluence. Cells were fixed with 3% paraformaldehyde (Science Services), permeabilized with 0.1% saponin (AppliChem) in PBS (wash buffer), and blocked with 3% bovine serum albumin (Sigma) in wash buffer. Wild-type cells were stained with mouse anti-ubiquitin (FK2, Enzo) and rabbit anti-p62 both diluted 1:500 in blocking buffer and subsequently with anti-mouse Alexa 546 and anti-rabbit Alexa 488 (Invitrogen). STG-p62 cells were stained only with mouse anti-ubiquitin followed by anti-mouse Alexa 546 as above. Cover slides were mounted with Dapi Fluoromount G (Southern Biotech) and imaged under a LSM710 microscope (Carl Zeiss).

For the experiment shown in Fig 6C, STGp62 wild-type and LIR mutant Hap1 cells were seeded and treated as above except that treatment was performed for 90 min, and staining was performed with mouse anti-ubiquitin FK2 antibody diluted to 1:1,000 and anti-mouse Alexa 546 diluted to 1:2,000. The laser intensity of the 488 laser was kept constant for all conditions and samples.

Images were analyzed using ImageJ with an automated procedure applying a fixed threshold for every channel, subtracting the area of the nuclei and analyzing all particles with intensity above the threshold.

## Fluorescence correlation spectroscopy

Cells were seeded on 10-well glass-bottom CellView Slides with TC surface (Greiner Bio-One). STG-p62 cells were seeded 1–2 days before imaging to reach no more than 50% confluence. For transfection of the GFP₁₋₅ ruler constructs, wild-type Hap1 cells were seeded directly on top of the transfection mix using 0.3 μl Fugene and 0.1 μg plasmid DNA per well. 24 h after transfection or 24–48 h after seeding, the medium was replaced with imaging medium (DMEMgfp-2, Evrogen) supplied with 20 μg/ml Rutin (Evrogen), 10% FBS (Sigma), and GlutaMAX™ (Gibco). Cells were subjected to

FCS measurements on a Olympus iX71 inverted microscope coupled to a MicroTime 200 TCSPC system (PicoQuant) operated by the PicoQuant SymPhoTime software suite. The system was equipped with a UPLSAPO 60× Ultra-plan-apochromat water immersion objective (NA 1.2), environmental control (37°C, 5% CO₂, humidity), a 481-nm pulsed laser diode, a 30-μm pinhole, a 525/45 emission filter, and a PicoQuant PMA Hybrid detector. Before measurements, the light path was aligned and the correction collar on the objective adjusted for maximum counts. The confocal volume was calibrated using a low nanomolar Atto488 solution, and autocorrelation curves were fitted with a diffusion coefficient of Atto488 at 37°C of 535 μm²/s. The Kappa value (ratio of axial and focal radii $z_0/w_0$) and the confocal volume ($V = p^{3/2} w_0^2 z_0$) for the experiment shown corresponded to 4.76 and 0.397 fL, respectively. Several spots per field were selected within the cytoplasm of cells where a uniform fluorescence could be detected. For cells transfected with the constructs of the GFP₁₋₅, ruler cells with the lowest visible fluorescence were selected. Data were recorded for at least 30 s per point with an applied laser power of 1 μW after the dichroic and a pulse rate of 40 MHz. Autocorrelation curves were calculated and fitted with Fluctuation Analyzer 4G (Wachsmuth *et al*, 2015). Curves were de-trended with the minimal trend correction window size set to 0.160 Hz and fitted assuming two diffusing components with unconstrained diffusion and protein-like triplet states with a fixed correlation time of 100 μs ($R^2 \geq 0.9934$). For GFP₁, 22 independent measurements were analyzed, 19 for GFP₂, 43 for GFP₃, 46 for GFP₄, 50 for GFP₅, and 52 for STG-p62. For interpolation of STG-p62 data into the calibration curves, LOG(MW) and the predicted number of GFP subunits were calculated for each measured curve, and then, the data were averaged and plotted with the relative SDs on both axes.

## Purification of STG-p62 from Hap1 cells

STG-p62 Hap1 cells were grown in suspension at 37°C in a 3 l Wheaton spinner flask for 5 days. The Minimum Essential Medium Eagle (Sigma-Aldrich) was supplemented with 6 g NaHCO₃, 10% bovine serum (Thermo Fisher Scientific, Waltham), 1% Pluronic™ F-68 Non-ionic Surfactant (Thermo Fisher Scientific), 1% MEM Non-Essential Amino Acids (Thermo Fisher Scientific), 1% Gluta-MAX™ Supplement (Thermo Fisher Scientific), Penicillin–Streptomycin (5,000 U/ml; Gibco, Thermo Fisher Scientific). STG-p62 cells were harvested by centrifugation at 1,300 $g$ for 15 min at 4°C and washed three times with PBS. Pellets were flash-frozen in liquid nitrogen, resuspended in ice-cold lysis buffer (50 mM Tris pH 7.5, 150 mM NaCl, 1 mM DTT supplemented with complete protease inhibitors EDTA-free cocktail, Roche Diagnostics), and cleared by centrifugation at 13,000 $g$ at 4°C for 15 min. For purification of STG-p62, 50 μl of StrepTactin Sepharose High performance beads (GE Healthcare) was added to 4 mg total protein in supernatants and incubated for 2 h at 4°C. Beads were washed four times with lysis buffer. Elution was performed in 50 μl of 5 mM biotin in lysis buffer for 60 min at 4°C. The eluted protein was subsequently used undiluted in cluster formation assays.

## Mass spectrometry

Three confluent 15-cm² plates of wild-type or STG-p62 Hap1 cells were harvested by trypsinization for 3′ at 37°C, spun at 100 $g$ for

10 min at 4°C, and washed three times in ice-cold PBS. Samples were then resuspended in cold lysis buffer [50 mM Tris pH 7.5, 150 mM NaCl, 1 mM DTT supplemented with complete protease inhibitors EDTA-free cocktail (Roche Diagnostics)], and frozen in liquid nitrogen, to disrupt the membrane. Lysates were thawed, spun at 500 $g$ for 10 min at 4°C, and the protein concentration was measured by Bradford. For each pull down, 3 mg of the total lysates was incubated with 20 μl of GFP-TRAP beads (Chromotek) for 90 min at 4°C under gentle rotation. The beads were washed four times with lysis buffer and subjected to trypsin digestion for mass spectrometry analysis. Peptides were analyzed by a LC-MS/MS, and acquired spectra were searched against an *in silico* digested protein database consisting of the human proteome (Uniprot) and common contaminants using the MaxQuant software. Proteins were relatively quantified across the samples using the label-free quantitation (LFQ) algorithm of the MaxQuant software. For ratio calculation, LFQ intensities were $\log_2$-transformed and missing values were replaced with a fixed value (17.2).

**Microscopy-based protein–protein interaction assays**

Assays were performed as described previously (Wurzer *et al*, 2015). GST-tagged proteins were recruited to the surface of glutathione Sepharose 4B beads (GE Healthcare), while GFP-NBR1 was recruited to GFP-TRAP beads (Chromotek). Assays were performed under equilibrium conditions with 2 μM of the prey proteins in SEC buffer (25 mM HEPES pH 7.5, 150 mM NaCl and 1 mM DTT). Beads were imaged either under a spinning disk (Visitron) microscope or a LSM710 confocal microscope (Carl Zeiss) at 20× or 25× magnification, respectively. For the experiments shown in Fig 3K, the buffer was supplemented with 1 μM final $ZnCl_2$. For quantification, six lines per bead were drawn across each bead in ImageJ and the maximal gray value across the line was taken. Values were averaged for each sample within each replicate, and then among replicates. For the experiment shown in Fig 5A, a total of 84 beads were quantified for GST from two independent replicates, 86 for GST-LC3B, and 72 for GST-4xUb. For the experiment shown in Fig 5B and C, a total of 77 beads from three independent experiments were quantified for p62 + GFP, 81 for p62 + GFP-NBR1, 91 for PB1 + GFP-NBR1, 77 for p62ΔPB1 + GFP-NBR1, and 83 for p62 Chimera + GFP-NBR1.

**Analytical size-exclusion chromatography**

Approximately 200 μg of mCherry-2xFKBP-p62 was run in the presence or in the absence of equimolar concentration of the AP20187 Homodimerizer (Takara) on a Superose6 10/300 column (GE Healthcare).

**Cluster formation assays**

All cluster formation assays were performed in 384-well plates with glass bottom (Greiner Bio-one). Unless otherwise specified, cluster formation assays were performed employing 2 μM mCherry- and/or GFP-p62 and 5 μM GST-GFP-4xUb or the indicated variants in SEC buffer. Cluster formation assays with mCherry-2xFKBP-p62 were conducted in the presence of 4 μM final concentration of AP20187 (Takara). Where relevant, for the dilution of p62 variants at

different stock concentrations or for different dilutions of the same p62 variant, the molarity of NaCl in the buffer was adjusted in each sample to have 150 mM final, taking into account the dilution of p62 (in 500 mM NaCl). For the experiments shown in Fig 3A and B, 20 μM GST-4xUb, 26.6 μM GST-3xUb, and 40 μM GST-2xUb were employed as final concentrations, respectively, so that the total amount of ubiquitin moieties would be 80 μM in all samples, regardless of the chain length. For the experiments shown in Fig 3C and D, 2 μM mCherry-p62 was pre-incubated with 1 μM 4xUb followed by the addition of 4 μM streptavidin immediately prior to imaging. For the experiments shown in Fig 3G, 2 μM GFP-p62 was incubated with 20 μM of the indicated GST-Ub chains. Due to the heterogeneity of chain lengths which did not allow the calculation of a precise extinction coefficient, for K48- and K63-linked chains, the protein concentration was calculated as described above employing the extinction coefficient of GST-1xUbΔGG, assuming that the contribution of each additional mono-ubiquitin moiety would be almost negligible (e.g., for GST-1xUbΔGG ε = 44,350 $m^2/$mol, for GST-4xUb ε = 48,820 $m^2$/mol). For the experiments shown in Fig 3H and I, 2 μM mCherry-p62 was incubated with 5 μM GST-4xUb in the presence or absence of 20 μM ubiquitin (the molarity refers to the number of ubiquitin moieties, regardless of the length of the chains). For the experiment shown in Fig 3J, 2 μM mCherry-2xFKBP-p62 was incubated with 4 μM AP20187 (Takara), 20 μM GST-4xUb, and 20 μM GFP or Ub chains (molarity referred to individual Ub moieties). For all assays, p62 variants were diluted to 1.1× in 10 μl SEC buffer. Pre-mixed proteins were diluted into p62 solution also at 1.1× concentration, and then, 1 μl of 11× cluster-inducing protein (e.g., GST-4xUb) was added immediately prior to imaging. Typical lag times from addition of GST-4xUb to start of imaging were below 1 min. Imaging was performed with a spinning disk microscope (Visitron) employing a 20× lens and a 14-bit Cool-SNAP HQ2 CCD camera (Photometrics). Samples were imaged for the indicated times with time points every 15 s, 30 s, or 1 min, depending on the number of samples acquired in parallel. Data were quantified as depicted in Fig EV1B using ImageJ (2.0.0-rc-49/1.51a build e01a259e5d). Briefly, the background was subtracted from each slide employing ImageJ's built-in rolling ball algorithm (radius 5px). Slices were subsequently thresholded with typical threshold values ranging from 250 to 500. The accuracy of thresholding was manually validated (see Fig EV1B) on the positive and negative controls and kept constant for all samples recorded in the same experiment. For p62 titration experiments, thresholds had to be readjusted from sample to sample due to the difference in fluorescence intensity of each sample. Each threshold was manually validated. After thresholding, the number of particles in each slide was analyzed with the function "Analyze -> Analyze Particles…" Only particles with sizes larger than 5px were picked. Circularity was not restrained. Particle analysis was automatized encoding the described workflow in an ImageJ macro (Code EV1).

"Count" and "Average Size" fields were taken into account for the data shown. "Average size" corresponds to the total area covered by particles (in $μm^2$) in each slice divided by the number of particles ("Count") in that slice. For the volume estimation, for each slice an average particle radius was calculated from the average size assuming a perfect circularity, then a spherical volume was calculated from that radius, again assuming perfect circularity. The total volume of particles in each slice (in $μm^3$) was obtained multiplying

the average particle volume (of that slice) with the particle count in that slice. Two independent fields of view per well (sample) were acquired, quantified separately, and subsequently averaged. The resulting average was considered as the readout from one experiment. Unless otherwise specified, at least three independent experiments per sample were imaged and analyzed for all the graphs shown.

### mCherry-hnRNPA1 liquid droplets formation

Droplets were formed as previously described (Lin *et al*, 2015). Briefly, the low-complexity domain (LCD) of human hnRNPA1 (amino acids 186–372) was cloned into pET-Duet-mCherry to generate 6xHis-TEV-mCherry-PSP-hnRNPA1$_{(LCD)}$. A PreScission Protease (PSP) site was included to potentially cleave the His-TEV-mCherry tag off. The protein was expressed in *E. coli* Rosetta pLysS o/n at 18°C in the presence of 0.1 mM IPTG and purified on a HisTRAP (GE Healthcare) column as described above. The eluted protein was further subjected to size-exclusion chromatography employing a Superdex 75 10/60 column (GE Healthcare) equilibrated with SEC buffer (25 mM Hepes pH 7.5, 150 mM NaCl, 1 mM DTT). Liquid droplets were formed diluting the protein to 6 μM final concentration in SEC buffer at 37.5 mM NaCl final molarity at RT in 384-well plate wells coated with BSA.

### Fluorescence recovery after photobleaching

*In vitro*: Clusters were formed as described above for 30 min in 384-well plates previously coated with 1 mg/ml BSA for 30 min at RT. Clusters and hnRNPA1 droplets sedimented on the bottom of the well were bleached with a 488-nm laser and imaged with a LSM710 confocal microscope (Carl Zeiss) equipped with a 63x/1.4 Plan-Apochromat oil-immersion lens. For quantification, a region of interest (ROI) was drawn within each bleached area and the average fluorescence intensity within the ROI was measured throughout the time lapse. The intensity of each particle before bleaching was set as 100%, while the intensity of the first time point after bleaching was set as 0%. All other intensities were normalized to these values.

*In vivo*: Acquisition was performed with a Zeiss Axio Observer Z1 inverse microscope equipped with a Plan-Apochromat 63×/1.4 Oil DIC objective, an EM-CCD back-illuminated evolve EM512 camera, a Yokogawa CSU-X1-A1 Spinning Disk Unit, an environmental control system, and the VisiView 3.3.0.6 software. FRAP was performed with Hap1 cell expressing endogenously tagged GFP-p62 (clone 3D). Defined areas of 20 μm were photobleached using a 488-nm laser at 50% laser power for 1 ms per pixel along 30 μm on the Z-axis. Every field was imaged as a Z-stack spanning the whole thickness of the cell. Time points were taken every 5 s over a total time of 205 s. Fiji (ImageJ) was used for quantification; Z-stacks were projected in single pictures as standard deviation Z projection, pictures were assembled in time-lapse stacks, and the fluorescence intensities of the very same dots in every slice were quantified. Fluorescence intensities at every time point were related to respective initial intensities at the pre-bleaching time point as percentages. FRAP curves were quantified measuring mean fluorescence intensity in the bleached region. The highest fluorescence intensity of pre-bleaching value was set to 100%, and lowest fluorescence intensity of post-bleaching value was set to 0%. Recovery (*r*) at any following

time point ($i_x$) was calculated as a fraction of pre-bleach minus post-bleach delta as $r = (i_x - i_{0\%})/(i_{100\%} - i_{0\%}) \times 100$.

### Structured illumination microscopy

Clusters were formed as described above for at least 30 min. If present, the second fluorescent p62 molecule was added and incubated for another 30 min. For pre-mixed mCherry- and GFP-p62 particles, the proteins were incubated with GST-4xUb for 1 h. For merged GFP- and mCherry-p62 particles, clusters were formed separately for 30 min, mixed, and incubated for another 30 min; 7 μl of the reaction volumes was subsequently transferred to high-precision microscopy coverslips (1.5H), mounted on cover glasses, and sealed with nail polish. Samples were imaged with a Deltavision OMX system (Applied Precision).

### Electron microscopy

p62 filaments were obtained as previously described (Ciuffa *et al*, 2015). p62–ubiquitin clusters were formed incubating 0.5 μM filaments alone or in the presence of 2.5 μM GST-4xUb; 3.5 μl was sampled at the indicated times after the addition of GST-4xUb and transferred onto glow-discharged carbon-coated EM grids (EM grid, Cu/Rh 300 mesh, Plano). Samples were absorbed on grids for 30 s, and subsequently stained shortly with 2% uranyl acetate, followed by a second staining of 40 s. Excess of solution was blotted away, and grids were left drying for 30 min. Grids were imaged using an FEI Morgagni operated at 100 kV.

**Expanded View** for this article is available online.

### Acknowledgements

We thank Graham Warren, Oliver Daumke, Thomas Leonard, and Alwin Köhler for comments on the manuscript. We are also grateful to Lijuan Zhang for obtaining the GFP$_{1-5}$ ruler constructs and the VBCF Advanced Microscopy facility for technical assistance with FCS and SIM. We thank Bettina Zens, Eleonora Turco, and Martina Schuschnig for assistance with cloning, protein expression, and purification and Ivan Yudushkin for sharing reagents. S.M. is supported by an ERC grant (No. 646653), by the Austrian Science Fund (FWF) (No. P30401-B21), and by the EMBO Young Investigator Program.

### Author contribution

GZ, AS, AD, JR, ME, MS, RT, ST, and AKT designed and performed the experiments and interpreted the results. SM and CS supervised the study, designed the experiments, and interpreted the results. TP wrote the macro for data analysis. GZ and SM wrote the manuscript and drafted the Figures.

### Conflict of interest

The authors declare that they have no conflict of interest.

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
