## [Review Process File · The EMBO Journal]

p62 filaments capture and present ubiquitinated cargos for autophagy

Gabriele Zaffagnini, Adriana Savova, Alberto Danieli, Julia Romanov, Shirley Tremel, Michael Ebner, Thomas Peterbauer, Martin Sztacho, Riccardo Trapannone, Abul K. Tarafder, Carsten Sachse and Sascha Martens

Review timeline:

Submission date:	28 September 2017
Editorial Decision:	12 October 2017
Revision received:	19 December 2017
Editorial Decision:	2 January 2018
Revision received:	3 January 2018
Accepted:	3 January 2018

Editor: Andrea Leibfried

Transaction Report:

1st Editorial Decision

12 October 2017

Thank you for submitting your manuscript for consideration by the EMBO Journal. It has now been seen by two referees whose comments are shown below.

As you will see, the referees appreciate your findings. However, they also note that the proposed significance of the findings in a physiological relevant context is not supported by data. Furthermore, they think that some controls are missing and that some more insight into K48 chain-mediated effects as well as LC3B-mediated effects and physiological significance is needed.

Given the referees' positive recommendations, I would like to invite you to submit a revised version of the manuscript, addressing the comments of all reviewers, and especially the points noted above; thus:

- please include all requested controls
- please add more insight on K48 and LC3B effects
- please use more careful wording to avoid over-statements

I should add that it is EMBO Journal policy to allow only a single round of revision, and acceptance of your manuscript will therefore depend on the completeness of your responses in this revised version.

Thank you for the opportunity to consider your work for publication. I look forward to your revision.

 REFEREE REPORTS

Referee #1:

Martens and coworkers determined how p62 forms clusters in vitro. They demonstrate that expression of p62 and GST- or streptavidin-tagged 4xUb is sufficient to produce clusters. These clusters are solid-like structures rather than typical liquid droplets. Formation of these clusters is affected by the presence of ubiquitin, polyubiquitin, and LC3. Unexpectedly, the LIR in p62 is important for the formation of p62 clusters independently of LC3 binding.

Most of the results presented here are convincing and appropriately quantified. Although the idea that p62 and ubiquitin together is sufficient to form aggregates may not be truly novel, the detailed characterization would be useful for the field. However, there are several points that need to be clarified.

Major concerns:

1. The authors suggest that there is a crosstalk between functioning proteasomes and autophagosome formation based on the in vitro results using ubiquitin and LC3. This suggestion is too speculative owing to a lack of relevant data. It would be OK to discuss this issue in Discussion but not in the Abstract and the last part of Introduction. This reviewer recommends removing the sentences pertaining to this issue.
2. Fig. 1: The number and size of Ub- and p62-positive clusters are different. The authors speculate that this is due to a high background signal of GFP-Ub. However, the mentioned background signal is not evident in Fig. 1c. Furthermore, Fig. 1g shows that the kinetics of formation of p62 clusters is much faster than that of Ub clusters, indicating that these are not identical structures. Is there a p62 subpopulation that is quickly formed with a smaller amount of Ub?
3. Fig. 4a: The authors show that p62-4xUb clusters are not dynamic. This reviewer has some concerns/suggestions.
 - (1) The results of the FRAP assay are difficult to interpret. The authors tried to bleach the fluorescence of the two clusters in the center of the field together with a surrounding area (liquid). At time 0, the circular area was bleached. However, the fluorescent intensities of not only the p62-Ub clusters but also the surrounding liquid area were not completely recovered following which. This is strange because fluorescent proteins should move freely in the liquid phase. Even if the clusters were not dynamic, the fluorescent signal of the surrounding liquid area should have been recovered immediately.
 - (2) As another approach, the authors could bleach the entire liquid fluorescence by continuously targeting a small section of the liquid area. If the clusters are indeed solid, they would not be bleached.
 - (3) It would be ideal to include typical liquid droplets (e.g., hnRNPA1) as a control.
 - (4) As this is an in vitro study, the results may not always reflect in vivo situations. It is thus important to discuss potential discrepancy between the present results and those in previous in vivo FRAP analysis of LC3, p62, and ubiquitin clusters (e.g., PMID: 22017874, 23482084, 27442348, 28369861, 28380357). Some previous studies demonstrated a more dynamic nature of p62 structures in vivo, which may be due to additional proteins that are incorporated into these clusters in vivo, weakening the interaction between p62 and other molecules. Whatever the reason, the authors should be more careful about generalizing these results obtained in the in vitro system and may want to consider adding, in Discussion, the possibility that of p62 structures being more dynamic in vivo.

Minor concerns:

Given that GST forms a dimer and streptavidin forms a tetramer, is streptavidin more active in generating clusters than GST?

Referee #2:

Misfolded proteins that cannot be degraded by the ubiquitin-proteasome system are cleared by selective macroautophagy. This requires controlled aggregation and binding of autophagy adapters such as p62. Zaffagnini and colleagues aimed at understanding the underlying biochemical mechanisms. They reconstituted p62-mediated aggregate formation *in vitro* using a ubiquitinated model substrate and microscopy based readouts. They show that aggregate formation requires ubiquitin chains of at least 4 subunits on cargo that needs to dimerize (i.e. harbour at least two chains). In p62, aggregate formation requires the ubiquitin-binding and the dimerization domains. They further analyse the aggregates and show that they are solid (excluding fluid phase separation) and contain distinct p62 fibres. Fluorescence-based analysis of aggregates in cells reveals that p62 is oligomeric and forms solid aggregates with ubiquitin. Using again the *in vitro* assay they show that the aggregation reaction is stimulated by NBR1, which is known to cooperate with p62 and affected by soluble LC3B and unanchored ubiquitin chains. They conclude that they recapitulated the early steps of aggregate autophagy showing that p62 filaments capture ubiquitinated cargo by aggregation, and that this is linked to autophagosome formation and proteasome functionality.

This is an overall careful and elegant study that recapitulates the collection of ubiquitinated cargo for autophagy. In doing so, it addresses and partially answers an important and timely aspect of a key element in maintenance of protein homeostasis. The assay in principle is elegant as it allows following kinetics and thus stimulatory and inhibitory effect of individual components. Also the characterization of the aggregates with the preformed p62 fibres is very interesting and an important advancement, although fibre p62 fibre formation has been demonstrated before. The conclusions on the functional links to proteasome activity and autophagosome formation are however based on little evidence. Also other conceptual concerns need to be addressed.

Detailed comments:

1) Although recapitulating these events *in vitro* is a major step forward, there is a general concern that the aggregation has a trivial explanation. Mixing two factors that bind each other with bivalent interaction sites (because they are dimeric) will inevitably lead to aggregation. I agree that the formation of p62 fibres (and the nice characterisation) speaks for a regulated mechanisms of "cluster formation" that likely reflects the *in vivo* situation. The authors also show that the process depends on the PD1 domain in p62, but since this affects oligomerization of p62 it does not exclude the possibility of trivial aggregation. The important converse question is if any Ub-binding oligomeric entity with comparable affinity for ubiquitin could do the same job as p62 and lead to similar cluster formation as observed in the *in vitro* assay.

2) page 9, end of second paragraph: the authors conclude that p62-aggregation is negatively regulated by active proteasome, because it is affected by unanchored K48 chains. This is far fetched and does not belong in the abstract. This could have many explanations. The whole observation needs further clarification. There is no mechanistic explanation provided for this observation, as p62 binds K63 better than K48. Also, the authors show that K63 chains (on the cargo) support cluster formation better. Yet, free K48 inhibit cluster formation better. The authors imply that they resolve the aggregates but this cannot be explained by competition with ubiquitin binding site in p62, since K63 do not work better.

3) Likewise, the basis for the effect of LC3B added to the reaction is not obvious. How does free LC3B inhibit cluster formation, and what would be the benefit. Moreover, is there evidence that LC3B inhibits cluster formation in cells? The conclusion that this effect points to a functional link to autophagosome formation is difficult to follow and not supported by sufficient evidence.

We would like to thank the reviewers for their overall positive assessment of our work. Please find below our point-by-point reply to your individual comments and suggestions.

Referee #1:

Martens and coworkers determined how p62 forms clusters in vitro. They demonstrate that expression of p62 and GST- or streptavidin-tagged 4xUb is sufficient to produce clusters. These clusters are solid-like structures rather than typical liquid droplets. Formation of these clusters is affected by the presence of ubiquitin, polyubiquitin, and LC3. Unexpectedly, the LIR in p62 is important for the formation of p62 clusters independently of LC3 binding.

Most of the results presented here are convincing and appropriately quantified. Although the idea that p62 and ubiquitin together is sufficient to form aggregates may not be truly novel, the detailed characterization would be useful for the field. However, there are several points that need to be clarified.

Major concerns:

1. The authors suggest that there is a crosstalk between functioning proteasomes and autophagosome formation based on the in vitro results using ubiquitin and LC3. This suggestion is too speculative owing to a lack of relevant data. It would be OK to discuss this issue in Discussion but not in the Abstract and the last part of Introduction. This reviewer recommends removing the sentences pertaining to this issue.

- We agree with the reviewer. We have now removed the relevant statements from the abstract and introduction and have rephrased the results and the discussion to use more careful wording.

2. Fig. 1: The number and size of Ub- and p62-positive clusters are different. The authors speculate that this is due to a high background signal of GFP-Ub. However, the mentioned background signal is not evident in Fig. 1c. Furthermore, Fig. 1g shows that the kinetics of formation of p62 clusters is much faster than that of Ub clusters, indicating that these are not identical structures. Is there a p62 subpopulation that is quickly formed with a smaller amount of Ub?

- Thank you for pointing this out. Indeed, the sample pictures that we included in the main figures were processed to remove background similar as during the quantification process. We have now included an example of the unprocessed images in Figure EV1c where the background is evident. We also believe that the reviewer is right, there is possibly a population of small clusters, especially at initial stages of formation, for which we were not able to capture any Ub signal above background. It might for example be that in an extreme case two p62 filaments containing multiple mCherry proteins are cross-linked by a single GST-GFP-4xUb dimer.

3. Fig. 4a: The authors show that p62-4xUb clusters are not dynamic. This reviewer has some concerns/suggestions.

(1) The results of the FRAP assay are difficult to interpret. The authors tried to bleach the fluorescence of the two clusters in the center of the field together with a surrounding area (liquid). At time 0, the circular area was bleached. However, the fluorescent intensities of not only the p62-Ub clusters but also the surrounding liquid area were not completely recovered following which. This is strange because fluorescent proteins should move freely in the liquid phase. Even if the clusters were not dynamic, the fluorescent signal of the surrounding liquid area should have been recovered immediately.

- We are grateful to the reviewer for raising this comment. We have now repeated this experiment in a technical better way using a different microscope. Previously we were imaging the clusters at the bottom of the plate and the surrounding signal noticed by the reviewer was due to the signal coming from protein adsorbed onto the glass bottom. We have now repeated the experiments using an improved setup on a confocal microscope, which allowed us to obtain

higher quality data. In particular it allowed us to bleach and quantify only a part of the cluster. This resulted in the observation that the p62 within the clusters is indeed immobile but in contrast to the previous results it became evident that ubiquitin is highly mobile. We have replaced Figure 4a with the new data and changed our conclusions accordingly. Please see Figures 4a and EV4a.

(2) As another approach, the authors could bleach the entire liquid fluorescence by continuously targeting a small section of the liquid area. If the clusters are indeed solid, they would not be bleached.

- We agree with the reviewer in that this would be an interesting experiment. However, due to the large volume of our imaging chamber compared to the volume we are able to bleach, this experiment is technically not possible in our setup.

(3) It would be ideal to include typical liquid droplets (e.g., hnRNPA1) as a control.

- Thank you for this suggestion. We have now included in Figure 4a and EV4a FRAP experiments with a liquid droplet-forming mCherry-tagged version of the low-complexity domain of human hnRNPA1. As expected, the hnRNPA1 protein showed fast recovery.

(4) As this is an in vitro study, the results may not always reflect in vivo situations. It is thus important to discuss potential discrepancy between the present results and those in previous in vivo FRAP analysis of LC3, p62, and ubiquitin clusters (e.g., PMID: 22017874, 23482084, 27442348, 28369861, 28380357). Some previous studies demonstrated a more dynamic nature of p62 structures in vivo, which may be due to additional proteins that are incorporated into these clusters in vivo, weakening the interaction between p62 and other molecules. Whatever the reason, the authors should be more careful about generalizing these results obtained in the in vitro system and may want to consider adding, in Discussion, the possibility that of p62 structures being more dynamic in vivo.

- We completely agree with the reviewer. We have now included FRAP experiments performed on p62 puncta in cells expressing endogenously GFP-tagged p62 (Figure 4b). To our knowledge this has previously not been done. We find that the endogenous p62 is more dynamic in the puncta in cells but that the p62 proteins are still fairly stably associated with the structures. We discuss these and the previous results in the manuscript.

Minor concerns:

Given that GST forms a dimer and streptavidin forms a tetramer, is streptavidin more active in generating clusters than GST?

- Thank you for addressing this point. Although it might be difficult to establish a precise comparison between the two sets of experiments due to the different kinetics, we believe that the answer to the question is yes, since in the streptavidin experiment the total amount of Ub moieties is 1 μ M (4 μ M 4xUb chains on one streptavidin tetramer). When a GST-GFP-4xUb substrate was provided at the same concentration of Ub moieties (Figure 3f, 1 μ M GST-GFP-4xUb) no relevant aggregation was detected.

Referee #2:

Misfolded proteins that cannot be degraded by the ubiquitin-proteasome system are cleared by selective macroautophagy. This requires controlled aggregation and binding of autophagy adapters such as p62. Zaffagnini and colleagues aimed at understanding the underlying biochemical mechanisms. They reconstituted p62-mediated aggregate formation in vitro using a ubiquitinated model substrate and microscopy based readouts. They show that aggregate formation requires ubiquitin chains of at least 4 subunits on cargo that needs to dimerize (i.e. harbour at least two chains). In p62, aggregate formation requires the ubiquitin-binding and the dimerization domains. They further analyse the aggregates and show that they are solid (excluding fluid phase separation) and contain distinct p62 fibres. Fluorescence-based analysis of aggregates in cells reveals that p62 is oligomeric and forms solid aggregates with ubiquitin. Using again the in vitro assay they show that the aggregation reaction is stimulated by NBR1, which is known to cooperate with p62 and affected

by soluble LC3B and unanchored ubiquitin chains. They conclude that they recapitulated the early steps of aggregate autophagy showing that p62 filaments capture ubiquitinated cargo by aggregation, and that this is linked to autophagosome formation and proteasome functionality.

This is an overall careful and elegant study that recapitulates the collection of ubiquitinated cargo for autophagy. In doing so, it addresses and partially answers an important and timely aspect of a key element in maintenance of protein homeostasis. The assay in principle is elegant as it allows following kinetics and thus stimulatory and inhibitory effect of individual components. Also the characterization of the aggregates with the preformed p62 fibres is very interesting and an important advancement, although fibre p62 fibre formation has been demonstrated before. The conclusions on the functional links to proteasome activity and autophagosome formation are however based on little evidence. Also other conceptual concerns need to be addressed.

Detailed comments:

1) Although recapitulating these events in vitro is a major step forward, there is a general concern that the aggregation has a trivial explanation. Mixing two factors that bind each other with bivalent interaction sites (because they are dimeric) will inevitably lead to aggregation. I agree that the formation of p62 fibres (and the nice characterisation) speaks for a regulated mechanisms of "cluster formation" that likely reflects the in vivo situation. The authors also show that the process depends on the PD1 domain in p62, but since this affects oligomerization of p62 it does not exclude the possibility of trivial aggregation. The important converse question is if any Ub-binding oligomeric entity with comparable affinity for ubiquitin could do the same job as p62 and lead to similar cluster formation as observed in the in vitro assay.

- The reviewer raises a good point and we have been thinking along similar lines. Ultimately, the model we propose is that p62 oligomers/filaments become crosslinked by ubiquitinated structures containing more than one ubiquitin chain. Thus, the physico-chemical basis for the phenomenon is, as often in cell biology, simple. However, the process has considerable specificity as only longer ubiquitin chains work, it is positively regulated by S403 phosphorylation and NBR1, and it is negatively regulated by free K48-/K63-linked ubiquitin chains and LC3B via its interaction with the LIR motif. More direct regulatory mechanisms can certainly be uncovered in our system in the future.

We have performed a new series of experiments employing a p62 variant with an artificially-induced oligomerization activity (Figures 3j, EV3e-f and EV6c). This variant is also able to sustain clustering but then the regulatory effect of the free K48-/K63-linked ubiquitin chains is lost. We have also performed the converse experiment providing wild-type p62 with another dimeric substrate (GST-LC3B, see below), which supported cluster formation only extremely weakly (about 150 cluster/field after 60 minutes as compared to about 6000 clusters for GST-4xUb after 2 minutes). In addition, our unpublished results suggest that our purified GFP-NBR1 is more than monomeric and yet it does not support cluster formation with GST-4xUb on its own (Figure 5d-e and Figure EV5a).

2) page 9, end of second paragraph: the authors conclude that p62-aggregation is negatively regulated by active proteasome, because it is affected by unanchored K48 chains. This is far fetched and does not belong in the abstract. This could have many explanations. The whole observation needs further clarification. There is no mechanistic explanation provided for this observation, as p62 binds K63 better than K48. Also, the authors show that K63 chains (on the cargo) support cluster formation better. Yet, free K48 inhibit cluster formation better. The authors imply that they resolve the aggregates but this cannot be explained by competition with ubiquitin binding site in p62, since K63 do not work better.

- Thank you for this comment. We have now rephrased and toned-down our conclusions also about the difference in the inhibition by free K48- and K63 linked chains. In addition, we have performed further experiments to understand the molecular mechanism underlying this effect. We could now show that the Zinc-finger (ZZ) domain of p62 binds to K48- and K63-linked ubiquitin chains (Figure 3k). Since the ZZ domain is positioned close to the oligomerizing PB1 domain (Figure 1a) ubiquitin binding to it might sterically interfere with oligomerization and could thereby negatively affect aggregation. This is at this point of course not proven but we feel that we can at least carefully speculate about this potential mechanism of action. Consistently, aggregation of the artificially-oligomerizing mCherry-2xFKBP-p62 construct was not as severely influenced by unanchored ubiquitin chains as the wild-type protein (Figure 3j), strongly suggesting that they exert their effect via the PB1 domain.

3) Likewise, the basis for the effect of LC3B added to the reaction is not obvious. How does free LC3B inhibit cluster formation, and what would be the benefit. Moreover, is there evidence that LC3B inhibits cluster formation in cells? The conclusion that this effect points to a functional link to autophagosome formation is difficult to follow and not supported by sufficient evidence.

- We have conducted further experiments to gain insights into the mechanism of the LC3B inhibitory effect and the role of the LIR motif for puncta formation in cells. *In vitro*, cluster formation by mCherry-2xFKBP-p62 was also affected by the addition of LC3B and we therefore concluded that the LC3B effect does depend on the PB1 domain (unlike free K48- and K63-linked chains; see above). We could also show that, under the conditions tested, the LIR motif of p62 does not mediate binding to full-length p62 (Figure EV6e), nor to ubiquitin, even when we employ a protein with four LIR motifs in tandem (Figure EV6f). Furthermore, we have tested the effect of the LIR mutation on p62 and ubiquitin-positive puncta formation in cells. To this end, we introduced the LIR mutation into the endogenous locus of the GFP-tagged p62 (STG-p62) cells using CRISPR/Cas9 gene editing. The LIR mutant cells formed

less p62- and ubiquitin-positive puncta upon acute proteotoxic stress induced with Puromycin (Figure 6c).

2nd Editorial Decision

2 January 2018

Thank you for submitting your manuscript for consideration by the EMBO Journal. It has now been seen by the two original referees whose comments are enclosed. As you will see, both referees are broadly in favour of publication, pending satisfactory minor revision.

I would thus like to ask you to address referee #2's remaining criticism and to provide a final version of your manuscript. It would be good to also update your synopsis text given this referee's input.

Please let me know in case you have any questions.

Thank you for the opportunity to consider your work for publication. I look forward to your revision.

REFEREE REPORTS

Referee #1:

The authors have adequately addressed all the criticisms to the previous version of this manuscript.

Referee #2:

The authors have addressed this reviewers concerns, but raised new points with the changes made to their manuscript that need to be solved before publication.

1) In their title and abstract, the authors now speak of "phase separation" as the basis of cluster formation and for cargo capturing by p62. This is not supported by the data. According to all their data, p62 forms solid aggregates/filaments that do not dynamically exchange and do not have properties of liquids. It is also inappropriate to fundamentally change the main conclusion of a submitted manuscript during the revision without even mentioning it in the response. It would require extensive justification and argumentation in the response. However, in this case the data do not even support this conclusion. In fact, the authors conclude that the aggregates are solid further below in the text.

2) In Fig. 4, the authors use a new setup for their analysis of cluster dynamics. They confirm that p62 is stationary (consistent with point 1), but they not find that 4xUb-GFP is dynamic. This could make sense with p62 forming filaments through their oligomerization domains, and dynamically binding ubiquitinated cargo through their IDR that stick out of the filaments. Given the significance of the finding, however, this is not sufficiently discussed and worked out in the manuscript.

First, it raises concerns as to why this result is opposite to their previous finding. The authors give technical reasons. However, how do we now that the new result is true and not the previous? Can any of the two be confirmed in cells?

Second, the authors' data demonstrate that p62 cluster formation requires 4xUb-GFP. Yet, once formed, the new data suggest that ubiquitin has higher mobility than p62, arguing that polyUb is not part of the network of protein-protein interactions that stabilizes the cluster. Rather, it suggests that ubiquitinated proteins induce the formation of p62-p62 interaction that are then independently maintained. All this needs to be discussed in more detail, as these are the significant implications of this particular contribution.

Referee #1:

The authors have adequately addressed all the criticisms to the previous version of this manuscript.

We thank the reviewer for the positive evaluation of our work.

Referee #2:

The authors have addressed this reviewer's concerns, but raised new points with the changes made to their manuscript that need to be solved before publication.

1) In their title and abstract, the authors now speak of "phase separation" as the basis of cluster formation and for cargo capturing by p62. This is not supported by the data. According to all their data, p62 forms solid aggregates/filaments that do not dynamically exchange and do not have properties of liquids. It is also inappropriate to fundamentally change the main conclusion of a submitted manuscript during the revision without even mentioning it in the response. It would require extensive justification and argumentation in the response. However, in this case the data do not even support this conclusion. In fact, the authors conclude that the aggregates are solid further below in the text.

Following the reviewer's criticism we have changed back the title of our manuscript to its original title reading "p62 filaments capture and present ubiquitinated cargos for autophagy". We have also removed the term "phase separation" from the abstract and exchanged it for "clustering". However, we want to point out that the vast majority of the conclusion of the manuscript has not fundamentally changed. p62 and ubiquitinated substrates do spontaneously cluster but during the more careful FRAP analysis performed for the revision it turned out that the ubiquitinated substrate employed for the experiment displays significant mobility, while the p62 filaments do not. Hence, we thought that the term solid aggregate is no longer the best term to describe the assemblies.

2) In Fig. 4, the authors use a new setup for their analysis of cluster dynamics. They confirm that p62 is stationary (consistent with point 1), but they not find that 4xUb-GFP is dynamic. This could make sense with p62 forming filaments through their oligomerization domains, and dynamically binding ubiquitinated cargo through their IDR that stick out of the filaments. Given the significance of the finding, however, this is not sufficiently discussed and worked out in the manuscript.

Thank you for this suggestion, we now discuss on page 17 "Thus, the clusters exhibit an intermediate behavior between solid and liquid phases (Wu & Fuxreiter, 2016), with a filamentous scaffold provided by the oligomeric PB1 domain and a dynamic interaction with the ubiquitinated substrates provided by the UBA domain, which is connected to the PB1 domain by a disordered region. The individual ubiquitin – UBA domain interactions may be transient and therefore the ubiquitinated substrates are able to move within the clusters. The larger p62 filaments on the other hand may be engaged in multiple substrate interactions at any given time and therefore be much more static. In addition, their larger size may limit their diffusion within the clusters. It is also conceivable that the ubiquitinated substrates trigger the formation of p62 – p62 interactions that are subsequently independently maintained."

First, it raises concerns as to why this result is opposite to their previous finding. The authors give technical reasons. However, how do we now that the new result is true and not the previous? Can any of the two be confirmed in cells?

We would like to point out that also in our initial experiment we found higher recovery of ubiquitin in the bleached particles. However, during the revision we have employed a better microscopy setup which allowed us to bleach only a part of the clusters. In addition, it allowed us to measure the recovery of the protein at a z-position that was away from the glass surface, to which the protein was absorbed unspecifically. Thus, we are strongly convinced that the new result is correct because it was obtained with a technically more advanced setup. We agree with the reviewer that it will be interesting to follow the behaviour of the ubiquitinated

substrates in cells and we are currently working towards this. However, following endogenous ubiquitinated substrates in an unbiased manner and without interfering with the cluster formation is technically demanding and we believe outside the scope of the current manuscript.

Second, the authors' data demonstrate that p62 cluster formation requires 4xUb-GFP. Yet, once formed, the new data suggest that ubiquitin has higher mobility than p62, arguing that polyUb is not part of the network of protein-protein interactions that stabilizes the cluster. Rather, it suggests that ubiquitinated proteins induce the formation of p62-p62 interaction that are then independently maintained. All this needs to be discussed in more detail, as these are the significant implications of this particular contribution.

Thank you for pointing this out. Please see our reply to your comment above under 2).

3rd Editorial Decision

3 January 2018

Many thanks for sending your revised manuscript to us. I appreciate the introduced changes, and I am happy to accept your manuscript for publication in The EMBO Journal.

Corresponding Author Name: Sascha Martens

EMBOJ-2017-98308R